# Auto-regressive In-context Demonstration Selection

**Yunzhe Qi** [* 1]  **Sirui Chen** [* 1]  **Jiaru Zou** [* 1]  **Jingrui He** [1]

## Abstract

Effective demonstration selection is crucial for maximizing large language model (LLM) performance in few-shot in-context learning. Because of effects such as recency bias, the effectiveness of demonstrations depends heavily on their contextual relationship to the specific query and on the ordering in which they are presented, making demonstration selection a complex combinatorial problem. To address these two challenges, we introduce AUTOSELECT, a novel framework that formulates demonstration selection as an auto-regressive sequential decision process. At each step, AUTOSELECT embeds the query and previously selected demonstrations into matrix representations to preserve structural information, and a trainable policy model sequentially selects the next best exemplar. To navigate the factorial space of demonstration permutations, our framework formulates a Kullback-Leibler (KL)-regularized optimization problem, from which an optimal policy induces an optimal Plackett-Luce (PL) ranking over all possible demonstration sequences. We prove that minimizing a tractable policy-level cross-entropy (CE) loss provably bounds the worst-case discrepancy between our policy's induced PL ranking and the optimal one, enabling tractable prioritization of high-quality sequences. Empirically, AUTOSELECT outperforms existing heuristic and learning-based methods across nine diverse datasets, achieving up to an 11% improvement over the strongest baseline. Analytical studies and a case study further highlight AUTOSELECT's key properties, as well as its transferability and generalizability.

[*]Equal contribution  [1]University of Illinois Urbana-Champaign. Correspondence to: Yunzhe Qi <yunzheq2@illinois.edu>, Jingrui He <jingrui@illinois.edu>.

*Proceedings of the $43^{rd}$ International Conference on Machine Learning*, Seoul, South Korea. PMLR 306, 2026. Copyright 2026 by the author(s).

## 1. Introduction

Large language models (LLMs) have demonstrated remarkable capabilities across diverse applications, including complex reasoning and code generation (Azerbayev et al., 2024; Imani et al., 2023; Shao et al., 2024; Zou et al., 2025). A key driver of inference-time performance is *few-shot in-context learning* (ICL), which enables LLMs to adapt to new tasks during testing using only a small number of demonstrations (exemplars) in the prompt (Brown et al., 2020; OpenAI et al., 2024; Anthropic, 2024), delivering strong performance while remaining computationally efficient.

Previous studies suggest that the effectiveness of few-shot ICL depends critically on two factors: the *content* of the selected demonstrations and their *ordering* (Lu et al., 2022; Zhao et al., 2021; Zhang et al., 2023). Influences such as recency bias can disproportionately affect model outputs and task performance (Peysakhovich & Lerer, 2023; Yang et al., 2026), making careful control over both aspects essential. The interaction between these factors gives rise to *compositional effects*, transforming *query-dependent* demonstration selection (Zhang et al., 2022; Chen et al., 2024b) into a complex combinatorial optimization problem: an exemplar highly effective for one query can be irrelevant, or even detrimental, for a different query, and the same set of exemplars can yield vastly different outcomes depending on their ordering (Dong et al., 2024). Many methods approach this combinatorial challenge by framing demonstration selection as a ranking problem, scoring candidate exemplars individually based on heuristic criteria such as semantic similarity (Reimers & Gurevych, 2019; Izacard et al., 2022). While they capture marginal content relevance, these methods can overlook pairwise and higher-order dependencies among exemplars and their ordering (the *compositional effects*) (Ye et al., 2023). This omission can bottleneck inference-time ICL performance.

Given that an exhaustive search for the optimal ordered sequence remains computationally intractable, there is a clear need for a more holistic yet tractable selection process. To address this, we draw inspiration from the core mechanism of LLMs themselves: the **auto-regressive paradigm** (Radford et al., 2018; Brown et al., 2020). By framing demonstration selection as a sequential decision process, we construct high-quality demonstration sequences tractably.

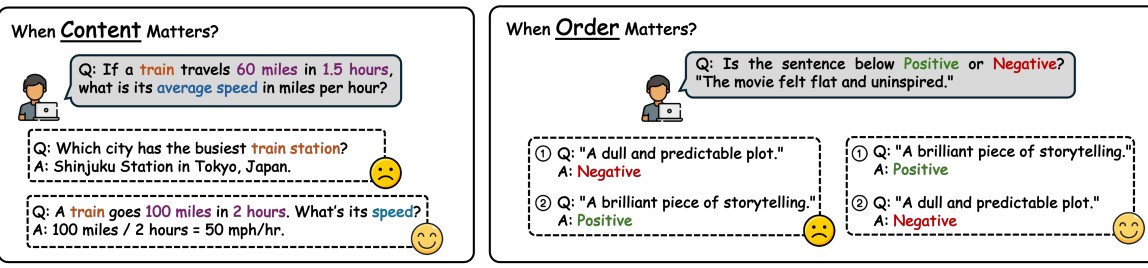

*Figure 1.* Sensitivity of in-context learning to demonstration content and order. **Left**: An example with relevant *content* (speed calculation) aids performance, while an irrelevant one does not. **Right**: The same set of examples can yield different outcomes based on their *ordering*.

Rather than retrieving isolated examples, our approach *composes* an effective demonstration sequence one step at a time, conditioned on the query and prior selections, much like an LLM generates a sentence token by token.

*Proposed Framework.* Motivated by the query-dependent and ordering-sensitive nature of few-shot ICL, we introduce AUTOSELECT, a novel auto-regressive framework built on a cohesive design for few-shot in-context demonstration selection. This problem is particularly challenging: even the simpler scenario of scoring an ordering by summing pairwise query-specific advantages reduces to the NP-hard Linear Ordering Problem (Martí & Reinelt, 2011). Thus, instead of attempting a direct and intractable search in this factorial space, AUTOSELECT re-frames the task as a dynamic, sequential decision process that mirrors how LLMs generate text. Our approach constructs a demonstration sequence one step at a time, conditioned on the input query and previous selections. This formulation is powered by a trainable policy model that operates on 2D matrix embeddings of exemplars and queries, a synergistic design that preserves internal token structure and captures complex inter-exemplar relationships. Meanwhile, AUTOSELECT also incorporates an adaptive stopping mechanism, allowing the model to dynamically learn the optimal sequence length for various queries. This unifies the critical aspects of demonstration selection, including content and ordering, into a single tractable framework.

*Theoretical and Empirical Insights.* To navigate the vast combinatorial space, AUTOSELECT employs a principled and theoretically grounded optimization procedure. We learn the demonstration selection policy by minimizing a tractable *policy-level cross-entropy (CE) loss*, guided by sequence-level rewards that inherently evaluate the *compositional effects* of both demonstration *content* and *ordering*. This is not a heuristic choice, since our theoretical analysis provides an intuitive guarantee: by modeling the problem as learning a Plackett-Luce (PL) distribution over demonstration-sequence rankings, minimizing this CE objective provably minimizes an upper bound on the discrepancy from the *optimal ranking*. This enables AUTOSELECT to efficiently prioritize high-quality demonstration sequences without exhaustive enumeration. Meanwhile, AU-

TOSELECT's effectiveness is also validated through comprehensive empirical evaluation. Across nine diverse tasks, AUTOSELECT consistently outperforms heuristic and learning-based baselines, achieving up to an 11% improvement over the strongest baseline. Notably, AUTOSELECT can deliver superior performance with considerably *fewer* demonstrations than top-$k$ retrieval, underscoring the practical value of optimizing demonstration quality rather than quantity alone. We further support these findings with extensive analytical studies on AUTOSELECT's behavior and demonstrate its generalization capabilities via a cross-task transferability case study.

## 2. Related Work

**Auto-regressive Paradigm.** The auto-regressive paradigm is foundational to sequential modeling and was significantly advanced by the Transformer architecture (Vaswani et al., 2017). This formulation intrinsically supports the success of LLMs (OpenAI et al., 2024; Anthropic, 2024; Grattafiori et al., 2024; Shao et al., 2024; Guo et al., 2025), and its effectiveness also extends to other modalities such as image and video generation (Weng et al., 2024; Tian et al., 2024). Crucially, recent adaptations of the auto-regressive paradigm to sequential decision-making, modeling reinforcement learning (RL) trajectories as sequences (Chen et al., 2021; Zheng et al., 2022; Lee et al., 2022), demonstrate its capability for complex sequential, ordered selection tasks. Thus, we formulate in-context demonstration selection as an auto-regressive problem and propose AUTOSELECT, which effectively accounts for both the input query and the crucial effect of exemplar ordering on demonstration selection.

**Few-shot In-context Demonstration Selection.** The performance of few-shot in-context learning is highly dependent on the chosen exemplars, including their quantity (Li et al., 2023), formatting (Jiang et al., 2020), and ordering (Zhao et al., 2021; Lu et al., 2022; Dong et al., 2024). A dominant paradigm for this task is *retrieval-based selection*, which generally treats the problem as a top-$k$ ranking task (Margatina et al., 2023). These methods include sparse retrieval methods such as BM25 (Robertson et al., 2009) and dense retrieval methods with learned semantic embeddings like Contriever (Izacard et al., 2022). Other retrieval methods

adopt the "select-then-rank" framework based on similarity and model-dependent scores (Peng et al., 2024). Meanwhile, *learning-based* approaches have been developed to capture richer inter-exemplar relationships. Some formulate the task as subset selection, using contrastive learning and Determinantal Point Processes to encourage diversity (Ye et al., 2023; Rubin et al., 2022). Others model the selection as a Markov Decision Process, training a policy with exemplar-level rewards (Zhang et al., 2022; Chen et al., 2024b). For example, Wang et al. (2025) formulate this as an RL problem to jointly optimize for both task relevance and exemplar diversity. Other research optimizes a single and static exemplar sequence shared across all queries for a given task (Min et al., 2022; Wu et al., 2024), where this task-level selection can help reduce inference-time overhead (Purohit et al., 2024; Halim et al., 2025). One category of these works formulates this as a subset selection problem to identify high-performing exemplars (Wu et al., 2024; Purohit et al., 2024; 2025), which can either consider exemplar ordering or remain order-independent. In contrast, AUTOSELECT constructs the demonstration sequence one step at a time, conditioning each choice on the full context of the query and previously selected exemplars. This enables AUTOSELECT to explicitly model crucial ordering and compositional effects using only sequence-level supervision, rather than potentially expensive exemplar-level supervision.

## 3. Preliminaries and Problem Definition

**Auto-regressive Paradigm.** With query $x$, an auto-regressive model (e.g., a language model) sequentially generates each output element by sampling from the conditional probability $\mathcal{P}(e_{i_t} \mid x, \tau_{<t})$, where $\tau_{<t} = (e_{i_1}, \ldots, e_{i_{t-1}})$ denotes the elements chosen before position $t$. The $t$-th element $e_{i_t}$ is then generated (e.g., sampled from the vocabulary), and the process terminates when a stopping criterion is met (e.g., reaching a maximum sequence length). This process naturally aligns with modern LLM generation (Li & Liang, 2021) and motivates our *"auto-regressive in-context demonstration selection"*.

**Auto-regressive In-context Demonstration Selection.** Suppose we have $N$ candidate demonstration examples (exemplars), represented by $\mathcal{E} := \{e_1, \ldots, e_N\}$, where each exemplar $e_i, i \in [N]$ refers to one query-answer pair. Here, given an input query $x$, our policy model $\pi_\theta$, parameterized by $\theta$, needs to select an *ordered sequence* of unique exemplars, containing at most $T$ elements ($T \leq N$): $\tau = (e_{i_1}, e_{i_2}, \ldots, e_{i_{|\tau|}})$, $|\tau| \leq T \leq N$. Our trainable policy model $\pi_\theta$ can be characterized by:

$$\pi_\theta(\tau \mid x) = \prod_{t=1}^{|\tau|} \pi_\theta(e_{i_t} \mid x, e_{i_1}, \ldots, e_{i_{t-1}}), \quad (1)$$

where $e_{i_t}$ refers to the $t$-th element in the generated sequence (trajectory), and the prefix $(x, e_{i_1}, \ldots, e_{i_{t-1}})$ is fed into the policy $\pi_\theta$ to choose the next exemplar, ensuring that

our selections adapt to both the query and previously chosen exemplars. The number of chosen exemplars $|\tau|$ can vary across input queries $x$. Throughout the paper, we use the terms *"sequence"* and *"trajectory"* interchangeably.

Under the combined system of the policy $\pi_\theta$ and a fixed task-solving LLM, we denote $\mathcal{P}_\theta(y \mid \tau, x)$ as the probability of the correct answer $y$, given query $x$ and exemplar sequence $\tau$. In this context, we aim to train the parameters $\theta$ of the policy model $\pi_\theta$, so that given an input query $x$, the policy-generated demonstration sequence (trajectory) $\tau \sim \pi_\theta(\cdot|x)$ maximizes the likelihood of the correct answer:

$$\max_\theta \ \mathbb{E}_{(x,y)} \Big[ \mathcal{P}_\theta(y \mid \tau, x) \Big], \quad (2)$$

which encourages the policy $\pi_\theta$ to assign higher probabilities to exemplar sequences that guide the LLM toward correct answers, tailored to different input queries.

## 4. Proposed Framework: AUTOSELECT

**Overview.** Fig. 2 illustrates the pipeline of the AUTOSELECT framework. (1) [Subsec. 4.1] We first embed exemplars $\mathcal{E}$ and input queries $x$ into matrix embeddings to preserve their structural information. For the input query in each training episode, we collect trajectories (i.e., exemplar sequences) of varying lengths using our policy model, and evaluate corresponding trajectory-level rewards. (2) [Subsecs. 4.2 and 4.3] With the collected trajectories and their rewards, we train our policy $\pi_\theta$ by minimizing our proposed policy-level cross-entropy (CE) loss, thereby reducing the discrepancy between $\pi_\theta$ and the optimal policy $\pi^*$. This refines the PL ranking induced by $\pi_\theta$ toward the optimal PL ranking induced by $\pi^*$, enabling $\pi_\theta$ to adaptively prioritize high-quality, query-specific exemplar sequences. Our pseudocode is in Algs. 1 and 2.

### 4.1. Policy Model and Trajectory Rollouts

**(I) Matrix Embedding.** To preserve structural information (e.g., the token sequence ordering that enables context-aware understanding and inter-token embedding relationships), we embed candidate exemplars $\mathcal{E}$ and the input query $x$ into individual embedding matrices, a strategy shown to effectively preserve structural information of text (Kim, 2014; Devlin et al., 2019; Khattab & Zaharia, 2020). Each embedding matrix is structured as follows: *rows* correspond to tokens from the exemplars or query (padded to a fixed length) and *columns* represent the embedding vectors of these tokens, obtained through a pre-trained embedding (e.g., GPT-2 embedding for our experiments). We slightly abuse the notation by also using $e \in \mathcal{E}$ to represent the exemplar *embedding matrix*. The input query $x$ will also be embedded into its *matrix representation* with this procedure.

**(II) Next-element Representation.** Recall that $e_{i_t}$ is the $t$-th element in the exemplar sequence as in Eq. 1. Follow-

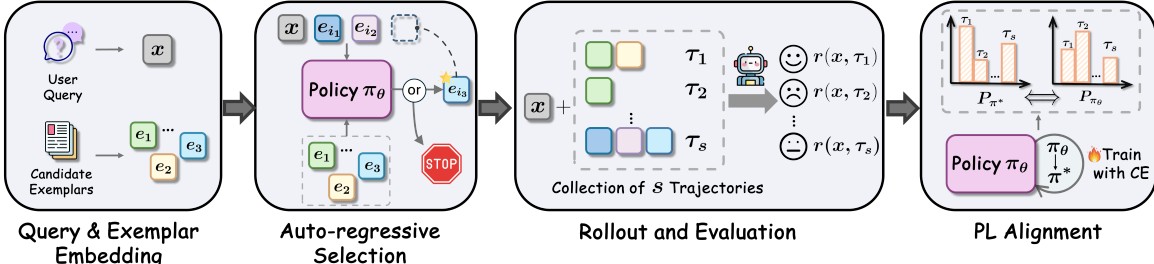

*Figure 2.* AUTOSELECT framework. We first embed candidate exemplars and the input query into matrix representations, then apply a trainable policy to sequentially select the next exemplar (or early stopping) conditioned on the query and prior selections. After evaluating collected exemplar sequences, our proposed CE loss is minimized to align the policy's induced PL ranking with the optimal PL ranking, thereby prioritizing high-quality exemplar sequences.

ing the auto-regressive paradigm, given an input query $\boldsymbol{x}$, denote the preceding $t-1$ chosen elements within the exemplar sequence as $\tau_{<t} = (\boldsymbol{e}_{i_1}, \boldsymbol{e}_{i_2}, \ldots, \boldsymbol{e}_{i_{t-1}})$. We apply a trainable encoding model $\phi(\cdot)$, such as a Transformer-based architecture (Vaswani et al., 2017), to generate the *matrix representation* $\boldsymbol{z}_t$ for the $t$-th selected exemplar $\boldsymbol{e}_{i_t}$. This is achieved by processing the concatenated sequence $(\boldsymbol{x}, \boldsymbol{e}_{i_1}, \ldots, \boldsymbol{e}_{i_{t-1}})$, and the resulting representation of the $t$-th element is

$$\boldsymbol{z}_t := \phi(\boldsymbol{x} \oplus \boldsymbol{e}_{i_1} \oplus \cdots \oplus \boldsymbol{e}_{i_{t-1}})$$

where $\oplus$ denotes the concatenation operation. $\boldsymbol{z}_t$ denotes the encoded matrix representation of the upcoming $t$-th element, which is shaped to match the padded token length and embedding dimension of exemplars and query matrices, enabling integration with matrix embeddings above.

**(III) Next-element Sampling.** We then select the $t$-th element by *sampling* from the distribution based on softmax-normalized distances (Mensink et al., 2013; Dong et al., 2016). Together, these yield the probability distribution of our trainable policy $\pi_\theta$ for selecting the $t$-th element:

$$\pi_\theta(\boldsymbol{e} \mid \boldsymbol{x}, \tau_{<t}) := \frac{\exp\left(-\gamma \cdot \|\boldsymbol{z}_t - \boldsymbol{e}\|_F^2\right)}{\sum_{\boldsymbol{e} \in (\mathcal{E} \setminus \tau_{<t}) \cup \{\boldsymbol{e}_{[\text{EOS}]}\}} \exp\left(-\gamma \cdot \|\boldsymbol{z}_t - \boldsymbol{e}\|_F^2\right)},$$
$$\forall \boldsymbol{e} \in (\mathcal{E} \setminus \tau_{<t}) \cup \{\boldsymbol{e}_{[\text{EOS}]}\},$$
$$(3)$$

where candidate choices at each step include all remaining candidate exemplars, along with a special *End-of-Sequence (EOS) signal* $\boldsymbol{e}_{[\text{EOS}]}$. This allows the model to dynamically determine the optimal length of the exemplar sequence. $\gamma > 0$ controls the sharpness of the distribution. Policy model implementation for experiments is detailed in Appendix A.2.

**(IV) Exemplar Sequence Reward Evaluation.** Policy $\pi_\theta$ is trained using sequence-level rewards, which are derived from the downstream task performance on sampled reference query-answer pairs $(\boldsymbol{x}, \boldsymbol{y})$. Given a query $\boldsymbol{x}$ and label $\boldsymbol{y}$, for an exemplar sequence $\tau$, we define its reward as the empirical measure of the sequence's effectiveness:

$$r(\boldsymbol{x}, \tau) := L\left(\boldsymbol{y}, \text{LLM}(\boldsymbol{x}; \tau)\right). \quad (4)$$

Here, $\text{LLM}(\cdot; \cdot)$ is the LLM response and $L(\cdot, \cdot)$ is the evaluation metric (e.g., accuracy). This reward serves as the training signal for maximizing our primary objective (Eq. 2). Note that in practice, the reward is *efficiently computed* using only a tiny batch of sampled reference examples per training episode (details in Appendix A.2.3). Thus, our training phase is a modest, one-time offline investment that yields a policy enhancing inference performance.

**(V) Full Trajectories & Sub-trajectories with Stopping.** To enable effective policy training, we need an informative collection of exemplar sequences (trajectories) $\mathcal{T}$, through a specialized rollout procedure from Alg. 2:

1. For a query $\boldsymbol{x}$, we generate $K$ rollouts, each capped at length $T$. At each step, the policy $\pi_\theta$ samples either a remaining candidate or the EOS signal $\boldsymbol{e}_{[\text{EOS}]}$.

2. When the EOS signal is selected, the current sub-trajectory is evaluated for its reward (Eq. 4) and stored (Alg. 2, line 8). To enable exploration of longer sequences while preserving dependencies, the rollout then continues by resampling a non-EOS exemplar.

3. The process concludes when the maximum length $T$ is reached, at which point the final *full trajectory* is also evaluated and stored in $\mathcal{T}$ (Alg. 2, line 14).

This efficiently populates $\mathcal{T}$ with both (i) *early-terminated* sub-trajectories and (ii) *full-length* trajectories from the same rollouts, preserving *dependencies* among prefixes. It enables the policy to effectively learn not only *which exemplars to select*, but also *when to adaptively terminate the sequence*.

### 4.2. RL Problem and PL Ranking of Trajectories

**Reinforcement Learning (RL) Problem.** To enable policy optimization without knowing the optimal demonstration sequence, we adopt a standard Kullback-Leibler (KL)-regularized RL objective below, as a practical surrogate for the learning objective in Eq. 2. Here, the commonly adopted KL-divergence term ensures stable policy optimization, by penalizing excessive deviation from a previous checkpoint

$\pi_{\text{old}}$ (Schulman et al., 2017; Rafailov et al., 2023; Chen et al., 2024a). For an input query $\boldsymbol{x}$ and the corresponding generated trajectories, our objective is to train the policy model $\pi_\theta$ by solving:

$$\max_{\pi_\theta} \mathbb{E}_{\tau \sim \pi_\theta(\tau|\boldsymbol{x})}\big[r(\boldsymbol{x}, \tau)\big] - \beta \cdot \mathbb{D}_{\text{KL}}\big(\pi_\theta(\tau \mid \boldsymbol{x})\|\pi_{\text{old}}(\tau \mid \boldsymbol{x})\big)$$

(5)

where $\tau = (\boldsymbol{e}_{i_1}, \ldots, \boldsymbol{e}_{i_{|\tau|}})$ refers to the generated trajectory as defined in Eq. 1, namely the chosen exemplar sequence of length $|\tau|$. Reward evaluation $r(\cdot, \cdot)$ is defined in Eq. 4, and the coefficient $\beta > 0$ controls the regularization intensity. It has been shown that the above optimization problem leads to a closed-form solution (Peters & Schaal, 2007; Rafailov et al., 2023), such that the *optimal policy*

$$\pi^*(\tau|\boldsymbol{x}) = \frac{1}{Z(\boldsymbol{x})} \cdot \pi_{\text{old}}(\tau|\boldsymbol{x}) \exp\big(\beta^{-1} r(\boldsymbol{x}, \tau)\big), \quad (6)$$

with $Z(\boldsymbol{x}) = \sum_{\tau'} \pi_{\text{old}}(\tau'|\boldsymbol{x}) \exp\big(\beta^{-1} r(\boldsymbol{x}, \tau')\big)$, where the intractable partition function $Z(\boldsymbol{x})$ is taken with respect to *all possible* trajectories.

**PL Ranking of Trajectories.** After generating a trajectory collection $\mathcal{T}$ and evaluating the trajectories' rewards, we need a principled way to *learn from their relative quality* and *prioritize high-quality exemplar sequences*. Thus, we formalize this using the PL model (Plackett, 1975; Luce et al., 1959), a standard probabilistic framework for modeling distributions over rankings based on utility scores.

Given a trajectory collection $\mathcal{T} = \{\tau_i\}_{i=1}^{|\mathcal{T}|}$ for query $\boldsymbol{x}$, for a permutation $\sigma$ of trajectory indices $\{1, \ldots, |\mathcal{T}|\}$, we have the optimal reward-based PL model induced by optimal policy $\pi^*$: $P_{\pi^*}(\sigma \mid \mathcal{T}, \boldsymbol{x}) := \prod_{i=1}^{|\mathcal{T}|} \frac{\exp(r(\boldsymbol{x}, \tau_{\sigma(i)}))}{\sum_{j=i}^{|\mathcal{T}|} \exp(r(\boldsymbol{x}, \tau_{\sigma(j)}))}$, where $\tau_{\sigma(i)}$ refers to the $i$-th ranked trajectory of permutation $\sigma$. This is equivalent to

$$P_{\pi^*}(\sigma \mid \mathcal{T}, \boldsymbol{x}) = \prod_{i=1}^{|\mathcal{T}|} \frac{\exp\left(\beta \log \frac{\pi^*(\tau_{\sigma(i)}|\boldsymbol{x})}{\pi_{\text{old}}(\tau_{\sigma(i)}|\boldsymbol{x})}\right)}{\sum_{j=i}^{|\mathcal{T}|} \exp\left(\beta \log \frac{\pi^*(\tau_{\sigma(j)}|\boldsymbol{x})}{\pi_{\text{old}}(\tau_{\sigma(j)}|\boldsymbol{x})}\right)},$$

and the equality follows by directly transforming Eq. 6 to derive the closed-form representation of $r(\cdot, \cdot)$.

---

**Why PL ranking?**

Our optimal policy $\pi^*$ (Eq. 6) assigns probabilities proportional to the exponential reward, $\pi^*(\tau \mid \boldsymbol{x}) \propto \exp(r(\boldsymbol{x}, \tau)/\beta)$, which matches the PL model's exponential scoring form. This makes the PL model naturally compatible with our goal of training $\pi_\theta$, so that its induced PL ranking matches the optimal one induced by $\pi^*$. Aligning with the optimal PL ranking trains $\pi_\theta$ to prioritize high-quality trajectories, thereby optimizing our main objective (Eq. 2).

---

**Bridging Policy and PL Ranking.** However, directly optimizing the discrepancy between PL models over permutations is infeasible, as there are $|\mathcal{T}|!$ possible permutations. To formulate a *tractable* policy training objective, we first theoretically bridge the PL ranking with policy optimization.

---

**Proposition 4.1** (Equivalence of Optimal PL Ranking and Optimal Policy). *A trainable policy $\pi_\theta$ is identical to the optimal policy $\pi^*$ (i.e., $\pi^* = \pi_\theta$) if and only if their PL ranking probabilities, $P_{\pi^*}(\sigma \mid \mathcal{T}, \boldsymbol{x}) = P_{\pi_\theta}(\sigma \mid \mathcal{T}, \boldsymbol{x})$, are equal for any possible trajectory collection $\mathcal{T}$. This relationship is also robust: by the invariance of the KL-regularized optimal policy to query-dependent reward shifts, the induced PL matching can faithfully capture the relative preferences among trajectories.*

---

The proof of Proposition 4.1 is in Appendix D. This equivalence motivates our strategy of training $\pi_\theta$ to match the properties of the optimal policy $\pi^*$, with the ultimate goal of achieving the optimal PL ranking $P_{\pi^*}$. Unfortunately, directly minimizing the universal discrepancy between $\pi^*$ and $\pi_\theta$ also remains infeasible: for a given input query $\boldsymbol{x}$, we only have access to a finite trajectory collection $\mathcal{T}$ rather than the full reward distribution. This motivates our practical policy-level cross-entropy (CE) loss, to be detailed in the next subsection.

### 4.3. Practical Objective: Policy-level CE Loss for PL Discrepancy Minimization

**Theoretical Intuition.** Here, since the partition function $Z(\boldsymbol{x})$ from Eq. 6 is intractable, we can compute the target probability distribution over trajectories induced by the optimal policy $\pi^*$ when restricted to a specific collection $\mathcal{T} = \{\tau_i\}_{i=1}^{|\mathcal{T}|}$. For any trajectory $\tau \in \mathcal{T}$, the probability conditioned on the collection $\mathcal{T}$ can be derived by renormalizing the expression (Eq. 6) over the collection $\mathcal{T}$:

$$\pi^*(\tau|\mathcal{T}, \boldsymbol{x}) = \frac{\pi_{\text{old}}(\tau|\boldsymbol{x}) \exp\left(\frac{r(\boldsymbol{x}, \tau)}{\beta}\right)}{\sum_{\tau' \in \mathcal{T}} \pi_{\text{old}}(\tau'|\boldsymbol{x}) \exp\left(\frac{r(\boldsymbol{x}, \tau')}{\beta}\right)}, \quad (7)$$

which provides the target relative likelihoods among the trajectories in the collection $\mathcal{T}$, with respect to the optimal policy $\pi^*$ (Eq. 6). Analogously, we can define the *conditional probability distribution* for our learnable policy

$$\pi_\theta(\tau|\mathcal{T}, \boldsymbol{x}) = \frac{\pi_\theta(\tau|\boldsymbol{x})}{\sum_{\tau' \in \mathcal{T}} \pi_\theta(\tau'|\boldsymbol{x})}. \quad (8)$$

With the above preliminaries, we motivate our training objective with Theorem 4.2 below, which shows that over a finite trajectory collection $\mathcal{T}$, the discrepancy between the PL rankings $P_{\pi^*}$ and $P_{\pi_\theta}$ can be minimized by instead reducing the conditional discrepancy between the policies $\pi^*$

---

**Algorithm 1** AUTOSELECT (One Training Episode)

---

1: **Inputs:** $T$, $\gamma$, number of trajectory rollouts $K$, embedded $\mathcal{E}$ and $\boldsymbol{e}_{[\text{EOS}]}$, replay buffer $\mathcal{B}$.

    { **Generating New Trajectories with $\pi_\theta$** }

2:   $\pi_{\text{old}} \leftarrow \pi_\theta$. Trajectory Collection $\mathcal{T} \leftarrow \emptyset$.
3:   Sample a reference query $\boldsymbol{x}$ and embed it.
4:   **for** $k \in \{1, \ldots, K\}$ **do**
5:      $\mathcal{T}_k \leftarrow \text{GenerateTrajectory}(\pi_\theta, \boldsymbol{x}, T, \gamma, k)$.
6:      $\mathcal{T} \leftarrow \mathcal{T} \cup \mathcal{T}_k$.
7:   **end for**

    { **Training with Instant Trajectories** }

8:   Compute CE loss $\mathcal{L}_{\text{CE}}$ (Eq. 10) with $\mathcal{T}$. Train $\pi_\theta$.

    { **Training with Replay Buffer** }

9:   Sample small batch $\widehat{\mathcal{B}} \subseteq \mathcal{B}$ from replay buffer $\mathcal{B}$.
10:  Calculate $\mathcal{L}_{\text{CE}}$ with $\widehat{\mathcal{B}}$ and update policy $\pi_\theta$.
11:  Update replay buffer $\mathcal{B} \leftarrow \mathcal{B} \cup \{(\boldsymbol{x}, \mathcal{T})\}$.

---

**Algorithm 2** GenerateTrajectory $(\pi_\theta, \boldsymbol{x}, T, \gamma, k)$

---

1: Initialize trajectory $\tau \leftarrow ()$, trajectory set $\mathcal{T}_k \leftarrow \emptyset$.
2: **for** $t = 1, \ldots, T$ **do**
3:    Get representation: $\boldsymbol{z}_t \leftarrow \phi(\boldsymbol{x} \oplus \tau_{<t})$.
4:    $p(\boldsymbol{e}) \leftarrow \text{Softmax}(-\gamma \|\boldsymbol{z}_t - \boldsymbol{e}\|_F^2)$,
                    $\forall \boldsymbol{e} \in (\mathcal{E} \setminus \tau_{<t}) \cup \{\boldsymbol{e}_{[\text{EOS}]}\}$.
5:    Sample $\boldsymbol{e}_{i_t} \sim \text{Categorical}(p(\boldsymbol{e}))$.
6:    **if** $\boldsymbol{e}_{i_t} == \boldsymbol{e}_{[\text{EOS}]}$ **then**
7:       Obtain reward for current $\tau$ (Eq. 4).
8:       Update set $\mathcal{T}_k \leftarrow \mathcal{T}_k \cup \{(\tau_{<t}, \boldsymbol{e}_{[\text{EOS}]})\}$.
9:       $p'(\boldsymbol{e}) \leftarrow \text{Softmax}(-\gamma \|\boldsymbol{z}_t - \boldsymbol{e}\|_F^2)$, $\forall \boldsymbol{e} \in (\mathcal{E} \setminus \tau_{<t})$.
10:      Re-sample: $\boldsymbol{e}_{i_t} \sim \text{Categorical}(p'(\boldsymbol{e}))$.
11:    **end if**
12:    Update trajectory $\tau \leftarrow (\tau_{<t}, \boldsymbol{e}_{i_t})$.
13:  **end for**
14: Obtain reward and update set $\mathcal{T}_k \leftarrow \mathcal{T}_k \cup \{(\tau_{\leq T}, \boldsymbol{e}_{[\text{EOS}]})\}$.
15: **return** Trajectory set $\mathcal{T}_k$

---

and $\pi_\theta$. The formal theorem and proof are provided and discussed in Appendix C.

> **Theorem 4.2** (PL Ranking Optimization via CE Loss Minimization (Informal)). *Given input query $\boldsymbol{x}$ and trajectory collection $\mathcal{T}$, let $\sigma$ be any permutation of trajectories in $\mathcal{T}$. The maximum absolute difference, between the probabilities assigned to $\sigma$ by the PL models of policies $\pi^*$ and $\pi_\theta$, can be bounded as*
>
> $$\max_\sigma \left| P_{\pi^*}(\sigma \mid \mathcal{T}, \boldsymbol{x}) - P_{\pi_\theta}(\sigma \mid \mathcal{T}, \boldsymbol{x}) \right|$$
> $$\leq \Phi\left(\mathcal{L}_{CE}^{\mathcal{T}}(\pi^*, \pi_\theta)\right), \qquad (9)$$
>
> *where $\mathcal{L}_{CE}^{\mathcal{T}}(\pi^*, \pi_\theta)$ is the CE loss conditioned on $\mathcal{T}$, between $\pi^*(\tau|\mathcal{T}, \boldsymbol{x})$ and $\pi_\theta(\tau|\mathcal{T}, \boldsymbol{x})$. $\Phi(\mathcal{L})$ decreases as $\mathcal{L}$ approaches its minimum, and $\Phi(\mathcal{L}) = 0$ when $\mathcal{L}_{CE}^{\mathcal{T}}(\pi^*, \pi_\theta)$ reaches its minimum, i.e., $\mathcal{L}_{CE}^{\mathcal{T}}(\pi^*, \pi^*)$.*

Theorem 4.2 suggests that the maximum PL ranking discrepancy of any permutation can be upper bounded by a function of the excess CE loss, which decreases as the CE loss approaches its minimum. Consequently, this indicates that minimizing the CE loss conditioned on the trajectory collection $\mathcal{T}$ can serve as a feasible training objective for learning the optimal PL ranking.

**Practical Training Objective: Minimizing Policy Discrepancy with CE Loss.** Motivated by the above insights, to train our policy $\pi_\theta$ to match the target distribution from the optimal policy $\pi^*$, we first define the training data $\mathcal{D}$ as a batch of query-trajectory-collection pairs $(\boldsymbol{x}, \mathcal{T})$ with corresponding rewards. For each pair, we can treat the target distribution $\pi^*(\tau|\mathcal{T}, \boldsymbol{x})$ from Eq. 7 as "soft labels" over the trajectories $\tau \in \mathcal{T}$. We then minimize the following CE loss between this target distribution and the distribution predicted by our learnable policy $\pi_\theta(\tau|\mathcal{T}, \boldsymbol{x})$:

$$\mathcal{L}_{\text{CE}}(\mathcal{D}) := -\frac{1}{|\mathcal{D}|} \sum_{(\boldsymbol{x}, \mathcal{T}) \in \mathcal{D}} \sum_{\tau \in \mathcal{T}} \left[ \pi^*(\tau|\mathcal{T}, \boldsymbol{x}) \cdot \log \pi_\theta(\tau|\mathcal{T}, \boldsymbol{x}) \right]$$
$$(10)$$

Minimizing $\mathcal{L}_{\text{CE}}$ helps align the learned policy with the optimal one, thereby prioritizing high-quality trajectories.

**Training with Instant Trajectories & Replay Buffer.** We apply *multi-episode* training for $\pi_\theta$. In each training episode, we receive an input query $\boldsymbol{x}$, generate a trajectory collection $\mathcal{T}$, and evaluate the corresponding rewards. (1) *Instant Trajectory Update*: Update $\pi_\theta$ (Alg. 1, line 8) by minimizing the CE loss (Eq. 10) computed on the current episode's collected trajectories $\mathcal{T}$ and their rewards. (2) *Replay-buffer Update*: Sample a small batch of past (query, trajectory-collection) pairs (Alg. 1, lines 9-11) and further update $\pi_\theta$ using the CE loss on this batch.

**Inference-time Demonstration Selection.** At inference time, the learned $\pi_\theta$ will generate exemplar sequences for test queries, following *Steps (I) to (III) in Subsec. 4.1*. For each query, its demonstration selection terminates upon selecting $\boldsymbol{e}_{[\text{EOS}]}$ or reaching maximum length $T$.

## 5. Experiments

**Experiment Settings.** We evaluate on nine datasets with diverse characteristics, including four commonly evaluated datasets (AGNews, Amazon, SST-2, Trec) in existing work on demonstration selection (Zhao et al., 2021; Zhang et al., 2022; Li et al., 2023), four BigBench (Srivastava et al., 2023) tasks (Winowhy, Epistemic_reasoning, Timedial, Hyperbaton) for testing LLMs' few-shot induction and reasoning capabilities, and the math-reasoning dataset AQuA (Ling et al., 2017). Following previous work on few-shot demonstration selection (Zhang et al., 2022; Wu et al., 2024), we set the maximum sequence length to 4, which is often treated as the standard *efficiency sweet spot* for moderate-sized LLMs, balancing significant performance gains with low computational overhead under our targeted few-shot in-context

*Table 1.* Comparison of AUTOSELECT with seven baselines (mean performance $\pm$ standard deviation over 3 seeds) and average ranks. Best results are shown in **bold** with dark green shading, and second-best are underlined with light blue shading. The final column reports AUTOSELECT's improvement over the best non-AUTOSELECT baseline when greedy-oracle is included; values in parentheses report the improvement when greedy-oracle is excluded.

| Task \ Method | Learning-free | | | Oracle | Learning-based | | | Ours | |
|---|---|---|---|---|---|---|---|---|---|
| | random | max-entropy | re-ordering | greedy-oracle | ActRL | CEIL | EASE | AUTOSELECT | Improvement (%) |
| AGNews | $0.767_{\pm0.027}$ | $0.774_{\pm0.035}$ | $0.773_{\pm0.040}$ | $\mathbf{0.848}_{\pm0.015}$ | $0.819_{\pm0.036}$ | $0.812_{\pm0.011}$ | $0.826_{\pm0.022}$ | $\underline{0.845}_{\pm0.005}$ | -0.4% (+2.3%) |
| Amazon | $0.911_{\pm0.008}$ | $0.938_{\pm0.000}$ | $0.939_{\pm0.007}$ | $\underline{0.943}_{\pm0.006}$ | $0.922_{\pm0.004}$ | $0.925_{\pm0.010}$ | $0.924_{\pm0.003}$ | $\mathbf{0.951}_{\pm0.004}$ | +0.8% (+1.3%) |
| SST-2 | $0.900_{\pm0.014}$ | $0.903_{\pm0.027}$ | $0.908_{\pm0.016}$ | $\underline{0.934}_{\pm0.001}$ | $0.916_{\pm0.021}$ | $0.912_{\pm0.061}$ | $0.922_{\pm0.016}$ | $\mathbf{0.946}_{\pm0.003}$ | +1.3% (+2.6%) |
| Trec | $0.217_{\pm0.025}$ | $0.277_{\pm0.009}$ | $0.303_{\pm0.049}$ | $0.370_{\pm0.018}$ | $0.283_{\pm0.040}$ | $\underline{0.375}_{\pm0.046}$ | $0.373_{\pm0.055}$ | $\mathbf{0.393}_{\pm0.023}$ | +4.8% (+4.8%) |
| Winowhy | $0.454_{\pm0.030}$ | $0.443_{\pm0.033}$ | $0.487_{\pm0.070}$ | $0.589_{\pm0.070}$ | $0.478_{\pm0.050}$ | $\underline{0.591}_{\pm0.037}$ | $0.580_{\pm0.005}$ | $\mathbf{0.657}_{\pm0.012}$ | +11.2% (+11.2%) |
| Epistemic_reasoning | $0.463_{\pm0.012}$ | $0.461_{\pm0.029}$ | $0.470_{\pm0.007}$ | $\underline{0.561}_{\pm0.021}$ | $0.482_{\pm0.039}$ | $0.546_{\pm0.043}$ | $0.532_{\pm0.012}$ | $\mathbf{0.601}_{\pm0.012}$ | +7.1% (+10.1%) |
| Timedial | $0.654_{\pm0.066}$ | $0.620_{\pm0.033}$ | $0.683_{\pm0.042}$ | $0.712_{\pm0.039}$ | $0.709_{\pm0.029}$ | $0.712_{\pm0.014}$ | $\underline{0.715}_{\pm0.011}$ | $\mathbf{0.738}_{\pm0.008}$ | +3.2% (+3.2%) |
| Hyperbaton | $0.516_{\pm0.037}$ | $0.508_{\pm0.026}$ | $0.516_{\pm0.015}$ | $0.551_{\pm0.026}$ | $0.573_{\pm0.041}$ | $\underline{0.610}_{\pm0.021}$ | $0.592_{\pm0.047}$ | $\mathbf{0.663}_{\pm0.011}$ | +8.7% (+8.7%) |
| AQuA | $0.348_{\pm0.014}$ | $0.346_{\pm0.024}$ | $0.355_{\pm0.008}$ | $\underline{0.374}_{\pm0.016}$ | $0.349_{\pm0.010}$ | $0.344_{\pm0.013}$ | $0.332_{\pm0.011}$ | $\mathbf{0.395}_{\pm0.002}$ | +5.6% (+11.3%) |
| Avg. Rank | 7.1 | 6.9 | 5.2 | $\underline{2.7}$ | 5.0 | 3.8 | 4.0 | **1.1** | – |

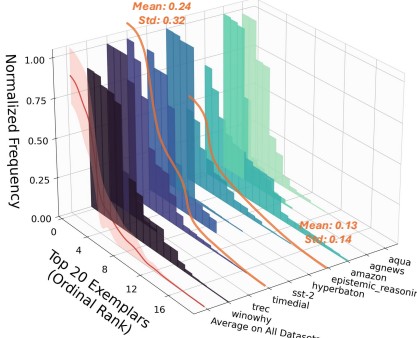

*Figure 3.* Top-20 exemplar selection frequencies across tasks.

learning settings (Purohit et al., 2025). For baselines, we include (1) heuristic learning-free methods: random, max-entropy, re-ordering; (2) an oracle-based method: greedy-oracle (Zhang et al., 2022) that selects the best candidate at each position via exhaustive enumeration, which is significantly more costly than AUTOSELECT and the other baselines; (3) three learning-based methods: Active Example Selection by RL (ActRL) (Zhang et al., 2022), CEIL (Ye et al., 2023), EASE (Wu et al., 2024). Qwen2.5-3B (Yang et al., 2025b) is applied as our task-solving LLM. Details are in Appendix A.

We first present the main empirical results: few-shot in-context learning experiments and a discussion of AUTOSELECT's properties (Subsec. 5.1), followed by complementary comparisons with order-independent baselines under various settings (Subsec. 5.1.1). We then present a case study demonstrating AUTOSELECT's transferability and generalizability, under both direct-transfer and adaptation settings (Subsec. 5.2). We also provide complementary experiments (e.g., results across different LLM families, hyperparameter studies, efficiency analysis, and inference-time analysis) in Appendix B.

### 5.1. Few-shot In-context Demonstration Selection

**Main Results.** In Table 1, AUTOSELECT generally outperforms strong baselines, benefiting from its effective policy design and the auto-regressive paradigm. The consistent

outperformance of the re-ordering method over the random baseline *empirically validates the importance of exemplar ordering*, a critical factor that AUTOSELECT is designed to exploit. While AUTOSELECT's improvement is marginal for saturated and less difficult tasks such as AGNews, AUTOSELECT can achieve substantial improvements on challenging ones, including Trec, four reasoning tasks, and the math dataset AQuA. CEIL generally outperforms EASE, particularly on challenging reasoning tasks, highlighting the importance of query-aware selection over fixed exemplars. While greedy-oracle achieves strong performance on certain tasks, it needs to exhaustively enumerate all exemplars and all the corresponding rewards, making it significantly more computationally expensive than AUTOSELECT and other baselines. However, greedy-oracle still overlooks exemplar compositional effects, leading to suboptimal performance.

**Properties.** From Fig. 3, AUTOSELECT can adaptively apply different selection strategies across tasks, while using only $\sim$ 3 exemplars on average (Fig. 10) with the EOS mechanism. This highlights its ability to capture task-dependent exemplar utility. AUTOSELECT also demonstrates strong performance across LLM families and scales (Appendix B.1), and yields consistent gains for increasing maximum sequence lengths up to $T = 16$ (Appendix B.2). Regarding efficiency, AUTOSELECT strikes a strong balance between *computational cost* and performance (Fig. 5; Appendix B.3), leveraging one-time offline policy training to enable efficient and effective demonstration selection.

### 5.1.1. COMPLEMENTARY COMPARISONS WITH ORDER-INDEPENDENT METHODS

*Table 2.* Comparison with order-independent and retrieval-based baselines.

| Method \ Task | Winowhy | Epi._reasoning | AQuA | Trec |
|---|---|---|---|---|
| Random | $0.454 \pm 0.030$ | $0.463 \pm 0.012$ | $0.348 \pm 0.014$ | $0.217 \pm 0.025$ |
| BM25 | $0.519 \pm 0.011$ | $0.497 \pm 0.003$ | $0.359 \pm 0.002$ | $0.364 \pm 0.004$ |
| Contriever | $0.538 \pm 0.027$ | $0.504 \pm 0.014$ | $0.357 \pm 0.005$ | $0.361 \pm 0.010$ |
| top-$k$ (Qwen2.5-3B Emb.) | $0.524 \pm 0.014$ | $0.501 \pm 0.013$ | $0.359 \pm 0.003$ | $0.375 \pm 0.014$ |
| EXPLORA | $0.552 \pm 0.017$ | $0.518 \pm 0.007$ | $0.351 \pm 0.005$ | $0.292 \pm 0.005$ |
| CASE | $0.534 \pm 0.044$ | $0.493 \pm 0.025$ | $0.374 \pm 0.006$ | $0.300 \pm 0.016$ |
| CEIL | $0.591 \pm 0.037$ | $0.546 \pm 0.043$ | $0.344 \pm 0.012$ | $0.375 \pm 0.046$ |
| AUTOSELECT | $\mathbf{0.657 \pm 0.012}$ | $\mathbf{0.601 \pm 0.012}$ | $\mathbf{0.395 \pm 0.002}$ | $\mathbf{0.393 \pm 0.023}$ |

In addition to learning-based CEIL, we also compare against

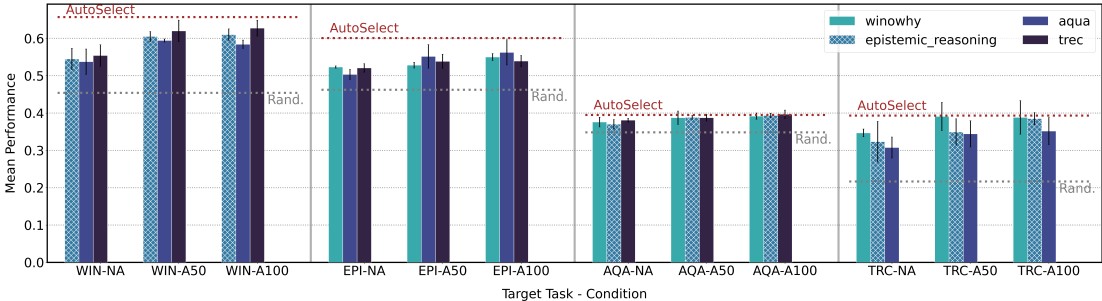

*Figure 4.* Transferability and generalization of trained policies from source tasks (legend) to target tasks (WIN: Winowhy; EPI: Epistemic_reasoning; AQA: AQuA; TRC: Trec) with Qwen2.5-3B. Horizontal lines "Rand." (Random) and AUTOSELECT denote reference performance levels (Table 1). X-axis: Target Task - Condition (NA: No Adaptation; A50/A100: 50/100 Adaptation Episodes).

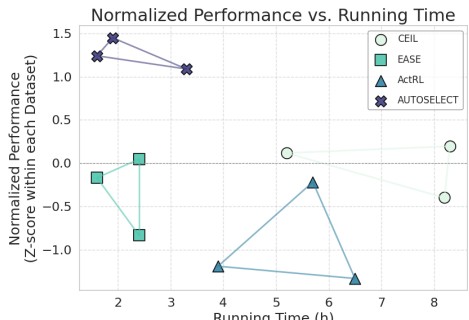

*Figure 5.* Efficiency analysis of normalized performance vs. overall running time (including training, testing, etc.). Each point represents a method-dataset pair on "AQuA", "Epistemic_reasoning", and "Winowhy". Performance is normalized within each dataset using Z-score scaling. AUTOSELECT consistently achieves a good balance between performance and runtime.

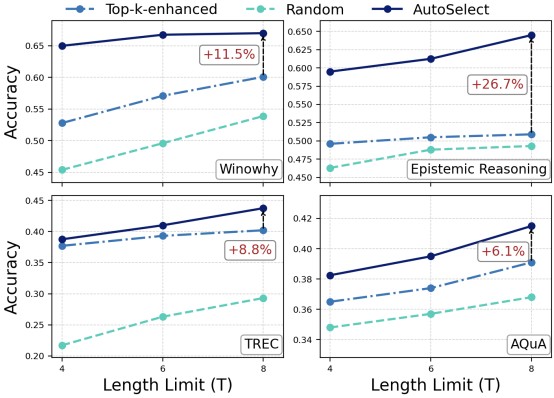

*Figure 6.* Comparison with the knowledge-enhanced top-$k$ method.

additional retrieval-based methods with pretrained embeddings: BM25 (Robertson et al., 2009), Contriever (Izacard et al., 2022), a top-$k$ method (Margatina et al., 2023) with native Qwen2.5-3B embeddings, as well as order-independent methods EXPLORA (Purohit et al., 2024) and CASE (Purohit et al., 2025). From Table 2, AUTOSELECT generally outperforms these baselines, underscoring the value of modeling exemplar interactions instead of relying on exemplar-level similarity or ranking scores alone. This indicates that exemplar selection guided by an auto-regressive policy can more effectively identify informative and task-relevant demonstrations, outperforming ordering-agnostic modeling and fixed similarity measures.

**Top-$k$ with Enhanced Knowledge.** We also compare with top-$k$-enhanced, a top-$k$ variant using enhanced knowledge and Qwen2.5-3B embeddings. Recall that rewards are computed on a validation set (Appendix A.2.3) to promote generalizable policy training. For fair comparisons, baselines requiring supervision signals (e.g., greedy-oracle and learning-based) similarly derive their supervision from the same validation set. In this context, top-$k$-enhanced leverages and selects exemplars from the union of the exemplar set $\mathcal{E}$ and the validation set. In Fig. 6, AUTOSELECT achieves stronger performance with considerably fewer demonstrations than

top-$k$-enhanced, especially for larger $T$, demonstrating that learned selection is key to better performance.

### 5.2. Transferability and Generalizability

We also include a case study on the transferability and generalizability of the trained policy $\pi_\theta$: (1) direct transfer to a new task without further training, and (2) transfer with a small number of adaptation episodes on the target task. From Fig. 4, when directly transferring trained policy models to different target tasks without adaptation, AUTOSELECT can already achieve better performance than simple heuristics. Moreover, a simple adaptation of 50 or 100 episodes (e.g., around 13 minutes on the "Trec" task for 50 episodes) can further improve its transferability. Notably, for "AQuA" and "Trec", the adapted policy can achieve performance comparable to task-specific optimization results (Table 1). Policies trained on "Winowhy" and "Epistemic_reasoning" tend to demonstrate strong generalization, as these tasks enable the policy to learn generalizable reasoning patterns. These results demonstrate AUTOSELECT's strong transferability, suggesting future extensions such as multi-task generalization, which we plan to explore in future work.

### 6. Conclusion

We propose and study auto-regressive in-context demonstration selection and introduce a novel framework, AU-

TOSELECT, to solve this problem. Utilizing a trained policy model, AUTOSELECT can effectively perform query-specific and ordering-aware exemplar selection for LLM few-shot in-context learning at inference time. Our theoretically grounded optimization procedure learns to approximate the optimal PL ranking from sequence-level rewards, efficiently bypassing exhaustive enumeration. AUTOSELECT empirically outperforms strong baselines across nine datasets, demonstrating robust generalization and adaptive selection. These results validate the effectiveness of the auto-regressive in-context demonstration selection paradigm and pave the way for future extensions such as cross-domain adaptation.

## Acknowledgements

This work is supported by IBM-Illinois Discovery Accelerator Institute - a new model of an academic-industry partnership designed to increase access to technology education and skill development to spur breakthroughs in emerging areas of technology. The content of the information in this document does not necessarily reflect the position or the policy of the Government, and no official endorsement should be inferred.

## Impact Statement

This paper introduces AUTOSELECT, an auto-regressive framework that advances LLM in-context learning by automating query-specific, ordered demonstration selection. This enhances LLM utility, accessibility, and potential transparency through understandable exemplars, while reducing manual effort. While our work has broad implications, we do not foresee significant negative impacts or societal concerns that warrant specific emphasis.

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

# A. Experiment Implementation Details and Complementary Discussions

## A.1. Baseline and Dataset Descriptions

Recall that we compare against seven baselines, categorized into three groups: (1) learning-free heuristic methods, (2) an oracle-based method, and (3) existing learning-based methods. They include:

- Type 1: *Heuristic Learning-free Baselines*

  - **Random**: This method chooses $T$ exemplars randomly as an exemplar sequence.
  - **Max-entropy**: It requires access to the logits of the task-solving LLM, and greedily selects examples that maximize classification entropy.
  - **Re-ordering** (Lu et al., 2022): To provide a controlled ablation on the *impact of sequence ordering*, this baseline operates on the exact same set of randomly sampled exemplars as the "random" baseline. Holding the content fixed, it then optimizes *only* the permutation (ordering) of these examples by selecting the order that maximizes classification entropy.

- Type 2: *Oracle-based Baseline*

  - **Greedy-oracle** (Zhang et al., 2022): At each step of exemplar selection, it greedily enumerates and evaluates all remaining candidates by appending each one to the current sequence and measuring its validation performance. For example, in a 5-shot scenario with a pool of 100 exemplars, once 4 have been chosen, it evaluates all 96 remaining candidates, requiring 96 separate validation runs. More generally, to validate exemplar sequences, greedy-oracle needs to query the task-solving LLM on the validation set for every possible combination of candidate exemplars at each selection step, which incurs a dramatically higher computational cost than other methods. This exhaustive enumeration is performed at every selection step: 100 runs for the first exemplar, 99 for the second, and so on through the fifth, making it significantly more computationally expensive.

- Type 3: *Existing Learning-based ICL Demonstration Selection Methods* [1]

  - **Active Example Selection by RL (ActRL)** (Zhang et al., 2022): It models the exemplar selection as a Markov Decision Process (MDP) and selects the exemplars with a Deep Q-network (DQN).
  - **CEIL** (Ye et al., 2023): It addresses in-context example selection by framing it as a subset selection task, employing Determinantal Point Processes (DPPs) to model the interplay between a given input and the in-context examples, with a contrastive learning objective.
  - **EASE** (Wu et al., 2024): It uses hidden embeddings from a pre-trained language model to represent ordered exemplar sequences and applies a neural bandit algorithm to optimize sequence formulation for each task, rather than performing query-aware exemplar selection.

We also provide dataset descriptions and exemplary query-answer pairs in Table 3.

## A.2. Implementation Details: Exemplar Sequence Generation with Vision Transformer (ViT)-based Policy Model, EOS Signal Instantiation, and Reward Evaluation

In this subsection, we provide instantiation details for our policy model architecture (Appendix A.2.1), implementation details of the EOS signal (Appendix A.2.2), as well as our reward evaluation procedure (Appendix A.2.3).

### A.2.1. VIT-BASED POLICY MODEL

**Policy Model Architecture and Intuition.** We instantiate our trainable encoding model $\phi(\cdot)$ and policy $\pi_\theta$ using a Vision Transformer (ViT) (Dosovitskiy et al., 2021), a natural choice for our auto-regressive selection process over 2D matrix embeddings. The pipeline instantiation is illustrated in Fig. 7. Recall that unlike methods requiring flattened vector inputs, our matrix representation preserves the vital token-level sequential structure of each exemplar. The ViT is uniquely suited to process our 2D structured matrix representations, allowing it to capture both internal token relationships and inter-exemplar dependencies.

---

[1]We omit empirical comparisons with an existing method (Chen et al., 2024b), due to the lack of an official public code implementation from the authors, and instead include the discussion in our Related Work section.

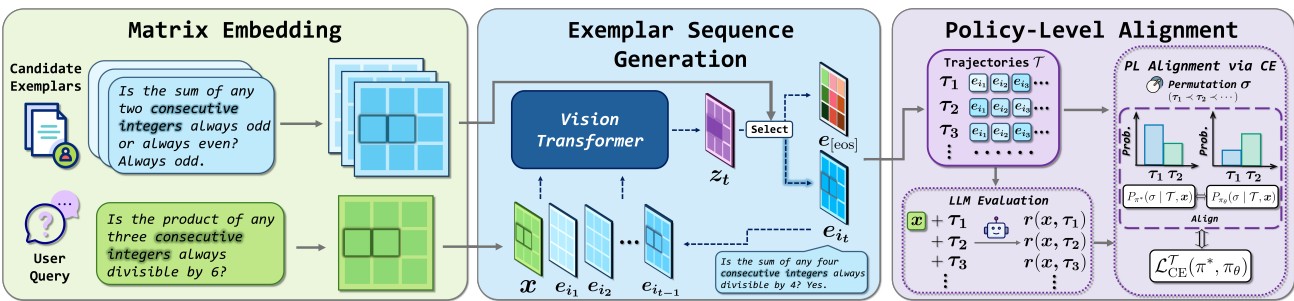

*Figure 7.* AUTOSELECT framework for auto-regressive demonstration selection. Queries and exemplars are first transformed into 2D matrix embeddings to preserve token-level structure. A ViT-based policy then auto-regressively processes the query and prior selections to generate a contextual representation, $z_t$, which guides the sequential selection of the next exemplar or an End-of-Sequence (EOS) signal. Finally, the generated sequences are evaluated by an LLM to obtain sequence-level rewards. These rewards are used to minimize a policy-level CE loss, aligning the policy's induced Plackett-Luce (PL) ranking with the optimal one to prioritize effective, query-aware sequences.

At each step $t$, the ViT processes the sequence of matrix embeddings for the query $x$ and preceding exemplars $(e_{i_1}, \ldots, e_{i_{t-1}})$ to synthesize the context into an output matrix representation $z_t$. This matrix acts as a dynamic prototype to guide the selection of the next exemplar, making this synergistic design a powerful and well-justified choice for our framework. Our implementation choice for the encoding model $\phi(\cdot)$ is also validated by an ablation study and visualization results in Appendix B.4. Our ViT-based encoding model generally achieves better performance, compared with a GPT-2 encoding model that has significantly more trainable parameters than our ViT-based model.

**Next-element Representation.** Then, at each step $t$ in the generation of an exemplar sequence, the policy needs to make a selection conditioned on the initial input query $x$ and all previously chosen exemplars, denoted by the prefix $\tau_{<t} = (e_{i_1}, e_{i_2}, \ldots, e_{i_{t-1}})$. The core of our policy model is a trainable ViT that processes the sequence of matrix embeddings corresponding to this context, $(x, e_{i_1}, \ldots, e_{i_{t-1}})$, to produce a contextual representation for the current step.

The ViT outputs a sequence of matrix embeddings, one for each input element. We take the final matrix embedding from this output sequence as a summary representation, $z_t$, which encodes the entire preceding context. This representation effectively serves as a dynamic "query" for selecting the next exemplar. Formally, $z_t$ is obtained as:

$$z_t := \text{ViT}(x \oplus e_{i_1} \oplus \cdots \oplus e_{i_{t-1}})[-1, :, :] \tag{11}$$

where $\oplus$ denotes concatenation along the sequence dimension, and the indexing $[-1, :, :]$ selects the last matrix embedding from the ViT's output tensor. The dimensionality of $z_t$ matches that of the input exemplar and query embeddings (i.e., padded token length $\times$ token embedding dimension).

**Next-element Selection.** With this context representation $z_t$, the policy selects the next element by matching $z_t$ against the embeddings of all available candidates. In this context, inspired by distance-based methods in vision tasks (Mensink et al., 2013; Dong et al., 2016), we use the squared Frobenius norm, $\|\cdot\|_F^2$, as a natural distance metric between these 2D matrix representations. These distances are then converted into a categorical probability distribution using a softmax function. The action space at step $t$ includes all exemplars not yet chosen, $(\mathcal{E} \setminus \tau_{<t})$, plus a special End-of-Sequence (EOS) signal, $e_{\text{[EOS]}}$, which allows the policy to terminate the sequence dynamically.

The complete probability distribution for our policy $\pi_\theta$ selecting the next element $e$ is given by:

$$\pi_\theta(e \mid x, \tau_{<t}) := \frac{\exp\left(-\gamma \cdot \|z_t - e\|_F^2\right)}{\sum_{e' \in (\mathcal{E} \setminus \tau_{<t}) \cup \{e_{\text{[EOS]}}\}} \exp\left(-\gamma \cdot \|z_t - e'\|_F^2\right)}, \quad \forall e \in (\mathcal{E} \setminus \tau_{<t}) \cup \{e_{\text{[EOS]}}\}. \tag{12}$$

The temperature parameter $\gamma > 0$ controls the sharpness of the distribution; a higher $\gamma$ makes the selection more deterministic by favoring candidates with the smallest distance, while a lower $\gamma$ encourages more exploration. The final element $e_{i_t}$ is then sampled from this distribution.

### A.2.2. EOS SIGNAL EMBEDDING

Analogous to the "EOS" token for generation termination in language modeling (Newman et al., 2020), we also formulate an "end-of-sequence" (EOS) embedding $e_{[\text{EOS}]}$ to serve as an ending signal for exemplar selection, when the policy model $\pi_\theta$ determines that the generated exemplar sequence is sufficient. Here, inspired by existing works on embedding initialization (Snell et al., 2017; Dobler & de Melo, 2023; Mundra et al., 2024), we set the embedding $e_{[\text{EOS}]}$ as the average exemplar embedding $e_{[\text{EOS}]} \leftarrow \lambda + \frac{1}{|\mathcal{E}|} \sum_{e \in \mathcal{E}} e$ with a small random perturbation $\lambda$. For the random perturbation $\lambda$ in the EOS signal embedding, we let $\lambda$ be a random matrix, whose elements are individually sampled from a zero-mean Gaussian distribution with standard deviation $0.01$. A complementary parameter study for $\lambda$ is also included in Appendix B.2.

### A.2.3. AGGREGATE METRIC REWARD

To obtain a fine-grained training signal when using possibly low-granularity evaluation metrics $L(\cdot, \cdot)$ (e.g., binary rewards) in our formulation (Eq. 4), we propose combining feedback by averaging the base metric outcomes over multiple input queries. Analogous techniques are commonly applied in reinforcement learning studies, particularly for sparse reward settings (Florensa et al., 2018). Specifically, given a small collection of query-answer pairs $\mathcal{D}_{\text{aggr}} = \{(\boldsymbol{x}_i, \boldsymbol{y}_i)\}_{i \in [|\mathcal{D}_{\text{aggr}}|]}$ sampled from the validation data (line 3, Alg. 1), we construct an aggregate query context $\bar{\boldsymbol{x}}$, defined as $\bar{\boldsymbol{x}} = \frac{1}{|\mathcal{D}_{\text{aggr}}|} \sum_{(\boldsymbol{x}, \boldsymbol{y}) \in \mathcal{D}_{\text{aggr}}} \boldsymbol{x}$, inspired by data augmentation techniques (Zhang et al., 2018). Then, the aggregated query $\bar{\boldsymbol{x}}$ is applied to generate the corresponding trajectory collection $\mathcal{T}$ (lines 4-7, Alg. 1). In this context, for each trajectory $\tau \in \mathcal{T}$ generated based on the aggregate query $\bar{\boldsymbol{x}}$, its reward is defined as:

$$r(\bar{\boldsymbol{x}}, \tau) := \frac{1}{|\mathcal{D}_{\text{aggr}}|} \sum_{(\boldsymbol{x}', \boldsymbol{y}') \in \mathcal{D}_{\text{aggr}}} L\big(\boldsymbol{y}', \text{LLM}(\boldsymbol{x}'; \tau)\big).$$

This formulation averages the base metric $L$ over the collection $\mathcal{D}_{\text{aggr}}$, yielding a smoother estimate of the trajectory $\tau$'s performance. The resulting aggregated query $\bar{\boldsymbol{x}}$, along with the generated trajectory collection $\mathcal{T}$ and their associated trajectory rewards, is subsequently used for policy training. In all our experiments, we set $|\mathcal{D}_{\text{aggr}}| = 5$, aggregating feedback from five individual queries, to enhance reward granularity while maintaining computational efficiency.

**Validation data.** For our experiments, the policy $\pi_\theta$ is trained using a reward signal derived from a validation set of query-answer pairs $(\boldsymbol{x}, \boldsymbol{y})$, which is kept separate from the candidate exemplar pool $\mathcal{E}$. This separation is a crucial methodological control, standard in the field (Zhang et al., 2022; Chen et al., 2024b; Wu et al., 2024), for two reasons. First, it prevents the policy from overfitting to a trivial *lookup* strategy. Second, by using the validation set *only* to generate a scalar reward signal, we compel the model to learn a generalizable selection skill.

### A.3. Experiment Implementation Details

For our few-shot in-context demonstration selection experiments, each task is associated with $|\mathcal{E}| = 100$ candidate exemplars and an additional 100 validation samples (query-answer pairs), which are distinct from the exemplar set $\mathcal{E}$. Additionally, we use another separate collection of 400 query-answer pairs as the test dataset, on which the performance of AUTOSELECT and all baselines is evaluated and reported in our results. The candidate exemplars, validation samples, and test samples are kept identical for AUTOSELECT and all baseline methods. For AUTOSELECT, we set the regularization coefficient to $\beta = 0.01$. A linear scheduler is applied to the temperature parameter $\gamma$, starting from $\gamma = 0.1$ and increasing linearly to $\gamma = 1$ over the first 200 episodes. The number of rollouts per episode is set to $K = 3$. Our policy model is trained over 400 episodes in a multi-episode training process. In each episode, we perform $K$ trajectory rollouts as indicated in Alg. 1. We set the replay buffer capacity to 50 and sample a small batch of size $|\widehat{\mathcal{B}}| = 10$ for each episode (line 9, Alg. 1), while updating the buffer using a FIFO (First-In, First-Out) strategy to discard outdated information.

To ensure consistency in input length, we pad all exemplars and input queries to a maximum of 320 tokens. For all our experiments, we use the AdamW optimizer (Loshchilov & Hutter, 2019) with the learning rate selected from $\{10^{-5}, 10^{-6}\}$. For our ViT-based policy model, input states are divided into square patches of size $32 \times 32$. The model consists of 4 Transformer blocks, each with an MLP dimension of 512, and 6 attention heads with a head dimension of 64. The output dimensionality of our ViT matches the shape of the query and exemplar embedding matrices, as described in Subsec. 4.1. All experiments are conducted on a Linux server with Intel Xeon CPU and NVIDIA A100 GPUs.

| Task | Descriptions & Query-output Examples |
|------|--------------------------------------|
| AGNews | A collection of news article titles and descriptions categorized into topics and used for text classification tasks. |
| | **Input**: No Need for OPEC to Pump More-Iran Gov TEHRAN (Reuters) - OPEC can do nothing to douse scorching oil prices when markets are already oversupplied by 2.8 million barrels per day (bpd) of crude, Iran's OPEC governor said Saturday, warning that prices could fall sharply. |
| | **Output**: Business. |
| Amazon | The Amazon dataset contains product reviews from Amazon, including ratings and review text, which are utilized for sentiment analysis and recommender system development. |
| | **Input**: This sound track was beautiful! It paints the scenery in your mind so well I would recommend it even to people who hate video game music! I have played the game Chrono Cross but out of all of the games I have ever played it has the best music! It backs away from crude keyboarding and takes a fresher step with great guitars and soulful orchestras. It would impress anyone who cares to listen. |
| | **Output**: Positive. |
| SST-2 | SST-2 includes sentence samples extracted from movie reviews, annotated with sentiment labels for sentiment analysis. |
| | **Input**: For those moviegoers who complain that "they don't make movies like they used to anymore." |
| | **Output**: Positive. |
| Trec | The Trec dataset involves fact-based questions labeled with semantic categories, designed for question classification evaluations. |
| | **Input**: How did serfdom develop in and then leave Russia? |
| | **Output**: Description. |
| Winowhy | The objective is to evaluate reasoning ability in answering Winograd Schema Challenge questions. |
| | **Input**: The city councilmen refused the demonstrators a permit because they feared violence. The 'they' refers to the city councilmen because The demonstrators advocated violence. |
| | **Output**: Correct. |
| Hyperbaton | The objective is to order adjectives correctly in English sentences. |
| | **Input**: Which sentence has the correct adjective order: a "small Iranian computer" b "Iranian small computer"? |
| | **Output**: a. |
| Epistemic_reasoning | The objective is to determine whether one sentence entails the next. |
| | **Input**: Premise: James understands that Charles thinks that three children hold a boy's arms down while another boy in a hat shoots a water gun at him. Hypothesis: Charles thinks that James understands that three children hold a boy's arms down while another boy in a hat shoots a water gun at him. |
| | **Output**: Non-entailment. |
| Timedial | The objective is to pick the correct choice for a masked (temporal) span given the dialog context. |
| | **Input**: Which phrase best fits the <MASK> span? Context: A: We need to take the accounts system offline to carry out the upgrade. But don't worry, it won't cause too much inconvenience. We're going to do it over the weekend. B: How long will the system be down for? A: We'll be taking everything offline in about two hours' time. It'll be down for a minimum of twelve hours. If everything goes according to plan, it should be up again by 6 pm on Saturday. B: That's fine. We've allowed <MASK> to be on the safe side. |
| | **Output**: 50 hours. |
| AQuA | The AQuA dataset consists of algebraic and arithmetic word problems in multiple-choice format, requiring both logical reasoning and numerical computation. |
| | **Input**: Two friends plan to walk along a 43-km trail, starting at opposite ends of the trail at the same time. If Friend P's rate is 15% faster than Friend Q's, how many kilometers will Friend P have walked when they pass each other? |
| | **Output**: 23. |

*Table 3.* Task descriptions and exemplary query-output pairs.

# B. Complementary Empirical Results

**Outline.** Due to strict page constraints in the main body, we include additional experiments in this Appendix section. The contents are organized as follows: (1) [Subsec. B.1] Experimental results of AUTOSELECT across diverse task-solving LLMs of various families and scales, as well as extra task types, demonstrating its broad compatibility. (2) [Subsec. B.2] Impact of hyperparameters on selection performance, with a parameter study highlighting performance trends, including trajectory length $T$, rollout count $K$, KL coefficient $\beta$, and EOS perturbation scale $\lambda$. (3) [Subsec. B.3] Discussion of sequence-length distributions and inference-time cost. (4) [Subsec. B.4] Backbone comparisons with a GPT-2-based variant of AUTOSELECT, validating our policy model architecture and implementation choices (Appendix A.2.1). (5) [Subsec. B.5] Qualitative examples of selected exemplar sequences and their potential correlations with input queries.

## B.1. Comparison across Varying Specifications

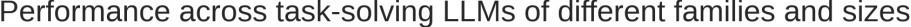

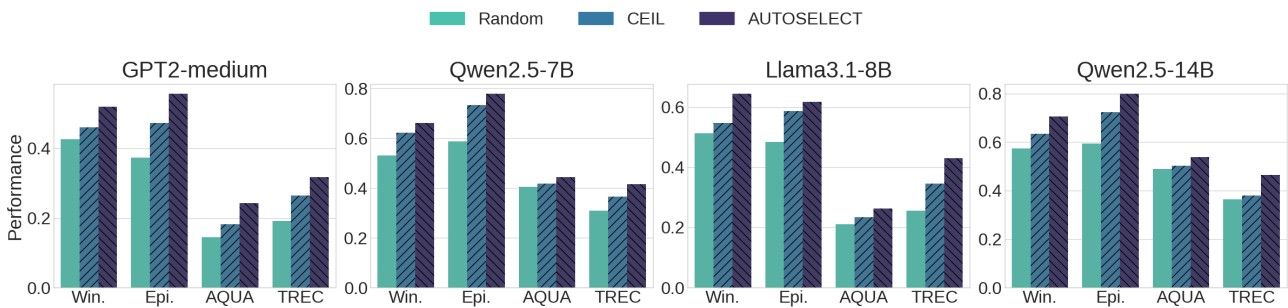

*Figure 8.* Performance comparison of exemplar selection methods across task-solving LLMs from various model families and scales. For abbreviations, "Win." and "Epi." respectively refer to the Winowhy and Epistemic_reasoning datasets.

To evaluate the effectiveness of AUTOSELECT across task-solving LLMs from different model families and scales, we compare performance across four datasets using GPT-2 Medium, LLaMA3.1 8B, and Qwen2.5 models with 7B and 14B parameters. The results, summarized in Fig. 8, provide a comprehensive view of how AUTOSELECT generalizes across varying model scales and architectures.

Across all models and datasets, AUTOSELECT consistently outperforms both random and CEIL baselines, demonstrating its adaptability and effectiveness, regardless of model size or architecture. Here, the improvements are particularly significant on more challenging reasoning tasks such as "Winowhy" and "Epistemic_reasoning", where precise semantic alignment and coherent exemplar context are critical. Compared to CEIL, AUTOSELECT's auto-regressive selection policy can more effectively capture the nuanced dependencies between queries and exemplars, especially when coupled with more capable language models. Another remarkable observation is that GPT-2 Medium and LLaMA3.1-8B show relatively poor performance on the "AQuA" dataset, even with improved exemplar selection. This is likely due to their limited math reasoning capabilities, which constrain the effectiveness of demonstration selection strategies.

**Additional results on generative tasks.** We evaluate AUTOSELECT on two generative tasks using Qwen3-8B (Yang et al., 2025a) with $T = 4$: XSum (Narayan et al., 2018), evaluated by ROUGE-L, and E2E-NLG (Dušek et al., 2020), with BLEU as the metric. AUTOSELECT consistently outperforms both BM25 and Random, suggesting that its benefit extends beyond classification-style tasks to open-ended generation settings.

*Table 4.* Generative-task evaluation with Qwen3-8B ($T = 4$).

| Dataset (Metric) | AUTOSELECT | BM25 | Random |
|---|---|---|---|
| XSum (ROUGE-L) | **0.2015** | 0.1886 | 0.1832 |
| E2E-NLG (BLEU) | **0.2614** | 0.2400 | 0.2361 |

**Additional results on Qwen3-8B.** We further evaluate AUTOSELECT on Qwen3-8B, a model post-trained with reinforcement learning. As shown in Table 5, AUTOSELECT consistently improves over both Random and Retrieval-based Top-$k$

selection across all evaluated tasks.

*Table 5.* Evaluation results on Qwen3-8B.

| Method \ Task | Winowhy | Epistemic_reasoning | AQuA | Trec |
|---|---|---|---|---|
| Random | 0.617 | 0.793 | 0.424 | 0.364 |
| Retrieval-based Top-$k$ | 0.661 | 0.827 | 0.451 | 0.412 |
| AUTOSELECT | **0.703** | **0.897** | **0.475** | **0.483** |

**Comparison with RDES.** We also implemented RDES (Wang et al., 2025) and adapted it to our setting for fair comparison. We run RDES with three random seeds and report the mean performance on Qwen2.5-3B. As shown in Table 6, AUTOSELECT outperforms RDES across evaluated tasks.

*Table 6.* Comparison with RDES on Qwen2.5-3B.

| Method \ Task | Winowhy | Epistemic_reasoning | AQuA | Trec |
|---|---|---|---|---|
| RDES | 0.577 | 0.525 | 0.375 | 0.382 |
| CEIL | 0.591 | 0.546 | 0.344 | 0.375 |
| AUTOSELECT | **0.657** | **0.601** | **0.395** | **0.393** |

## B.2. Effect of Hyperparameters on Selection Performance

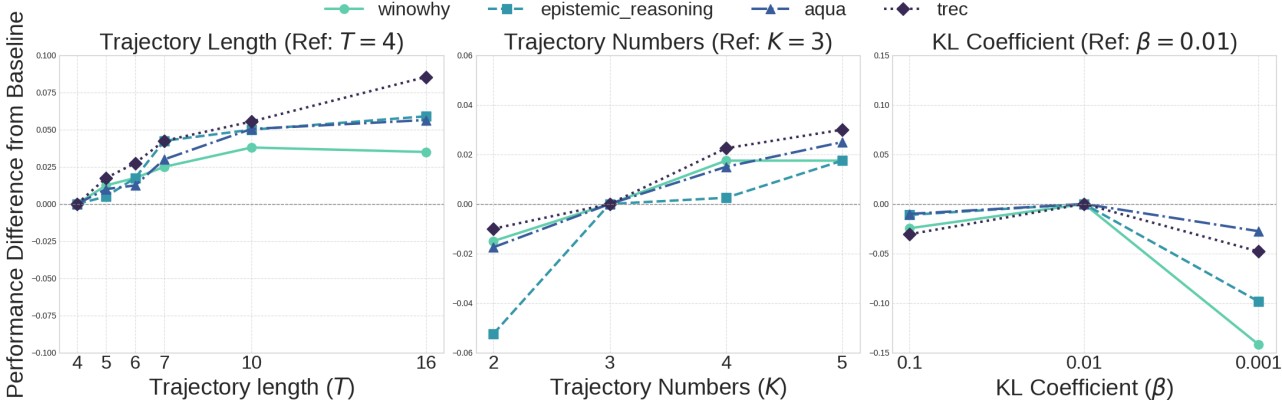

*Figure 9.* Parameter sensitivity analysis of AUTOSELECT. **Left:** Impact of trajectory maximum length $T$. **Middle:** Effect of the number of trajectory rollouts $K$. **Right:** Influence of KL regularization coefficient $\beta$. All results are reported as relative performance improvements over default settings ($T = 4$, $K = 3$ and $\beta = 0.01$).

In this subsection, we conduct a parameter study to analyze the effect of key hyperparameters in AUTOSELECT, including maximum trajectory length $T$, number of trajectory rollouts $K$, and KL regularization coefficient $\beta$. As shown in Fig. 9, results on four datasets are reported as relative performance improvements over the default settings used in our main experiments (Table 1).

As shown in Fig. 9 (left), increasing the maximum trajectory length $T$ from 4 to 7 consistently improves performance across four tasks, suggesting that longer trajectories generally offer richer contextual signals for the task-solving LLM. Additionally, Fig. 9 demonstrates that increasing the number of trajectory rollouts $K$ from 2 to 5 can also consistently improve performance across four tasks, with the most notable gains observed between $K = 2$ and $K = 4$. This indicates that generating more trajectories can enhance exemplar diversity and improve reward signal quality. Meanwhile, Fig. 9 (right) shows the sensitivity to $\beta$, which governs the KL constraint strength during policy updates. We find $\beta = 0.01$ consistently yields the best performance. Larger $\beta$ values can result in overly conservative updates, while very small $\beta$ values (e.g., 0.001) can cause policy optimization instability due to insufficient KL regularization.

**Perturbation Scaling** $\lambda$**.** Recall that we introduce a small random perturbation $\lambda$ to the end-of-sequence (EOS) signal embedding in our instantiation, which is detailed in Appendix A.2. For the random perturbation $\lambda$

| Task \ $\lambda$ | $\lambda = 0$ | $\lambda = 0.01$ | $\lambda = 0.05$ | $\lambda = 0.1$ |
|---|---|---|---|---|
| Winowhy | $0.641 \pm 0.018$ | $0.657 \pm 0.012$ | $0.647 \pm 0.038$ | $0.639 \pm 0.022$ |
| AQuA | $0.379 \pm 0.004$ | $0.395 \pm 0.002$ | $0.388 \pm 0.008$ | $0.391 \pm 0.011$ |

*Table 7.* Performance comparison on the Winowhy and AQuA tasks for different $\lambda$ values.

in the EOS signal embedding, we let $\lambda$ be a random matrix, whose elements are individually sampled from a zero-mean Gaussian distribution with standard deviation $\lambda$. Our empirical results suggest this perturbation is beneficial, with the best performance on both Winowhy (0.657) and AQuA (0.395) achieved when setting the noise scaling coefficient to $\lambda = 0.01$. We also investigate the sensitivity of the noise scaling coefficient $\lambda$ for our EOS signal embedding instantiation. From the results in Table 7, AUTOSELECT maintains stable performance across different choices of $\lambda$.

**Training and Inference Time Scaling with Larger Exemplar Pools.** For AUTOSELECT at inference time, the cost of each exemplar selection step scales linearly with the size of the candidate pool due to the policy's softmax computation. Training time is similarly affected during trajectory rollouts, while the subsequent gradient-update cost is not affected by exemplar pool size.

Empirically, selection cost remains modest even as the pool grows. The time increases from 6–7 ms (100 exemplars) to 31–34 ms (1000 exemplars); this non-proportional scaling suggests a constant base latency for state updates, alongside an approximately linear cost of candidate scoring. This is consistent with each auto-regressive decision requiring only a single forward pass over the current pool, so the linear complexity is $O(TN)$ in horizon ($T$) and pool size ($N$). The selection time (in seconds) for each exemplar is shown below.

*Table 8.* Selection time scales mildly with exemplar pool size.

| Method \ Task | Winowhy | Epistemic_reasoning | AQuA | Trec |
|---|---|---|---|---|
| AUTOSELECT (Pool Size: 100) | 0.006 | 0.006 | 0.007 | 0.006 |
| AUTOSELECT (Pool Size: 500) | 0.017 | 0.016 | 0.017 | 0.017 |
| AUTOSELECT (Pool Size: 1000) | 0.032 | 0.032 | 0.031 | 0.034 |

In practice, the dominant cost in our full system comes from LLM calls for reward evaluation, which are independent of the exemplar pool size. The additional 20–30 ms per selection when moving from 100 to 1000 exemplars is negligible compared to LLM inference time and comfortably fits within a single GPU budget. Thus, while our method is sequential over steps, the empirical results show that it remains computationally practical for large exemplar pools.

**Size and Quality of Validation Set.** AUTOSELECT mitigates sensitivity to noise or low-granularity rewards by using an Aggregate Metric Reward. This aggregation averages feedback over a small batch of validation samples to create a smoother and more stable training signal. In this case, we conduct further experiments by (1) changing the size of the validation set, as well as (2) imposing noise perturbations (zero-mean Gaussian noise) on the reward signals derived from the validation set. These results are shown below:

*Table 9.* Validation set size has limited impact.

| Method \ Task | Winowhy | Epistemic_reasoning | AQuA | Trec |
|---|---|---|---|---|
| AUTOSELECT (Validation Size: 100) | 0.657 | 0.601 | 0.395 | 0.393 |
| AUTOSELECT (Validation Size: 200) | 0.669 | 0.573 | 0.391 | 0.399 |
| AUTOSELECT (Validation Size: 400) | 0.640 | 0.596 | 0.388 | 0.384 |
| AUTOSELECT (Validation Size: 500) | 0.635 | 0.581 | 0.406 | 0.404 |

*Table 10.* Robustness to noisy rewards.

| Method \ Task | Winowhy | Epistemic_reasoning | AQuA | Trec |
|---|---|---|---|---|
| AUTOSELECT (No Noise) | 0.657 | 0.601 | 0.395 | 0.393 |
| AUTOSELECT (Gaussian Noise Std. = 0.01) | 0.650 | 0.587 | 0.381 | 0.388 |
| AUTOSELECT (Gaussian Noise Std. = 0.05) | 0.643 | 0.568 | 0.388 | 0.382 |
| AUTOSELECT (Gaussian Noise Std. = 0.1) | 0.628 | 0.550 | 0.373 | 0.368 |

From Tables 9 and 10, we see that: (1) Validation Size: Performance is relatively stable across different validation set

sizes, as we only sample a small batch of validation samples to derive each reward. This suggests that our method is sample-efficient and capable of extracting a robust policy without requiring large-scale validation data. (2) Noise Sensitivity: As expected, we see that performance can mildly degrade as noise increases. AUTOSELECT retains strong performance under low noise ($\sigma = 0.01$) and remains functional at moderate noise ($\sigma = 0.05$), confirming that the Aggregate Metric Reward design can effectively smooth out fluctuations to provide a reliable training signal.

**Aggregation Size.** By varying $|\mathcal{D}_{\text{aggr}}|$ across different values, we observe a consistent trend where increasing the aggregation size $|\mathcal{D}_{\text{aggr}}|$ generally improves performance. This supports the idea that averaging the reward over a batch of queries (Aggregate Metric Reward) helps reduce the variance of the reward signal, providing a more stable learning objective for the policy. We apply $|\mathcal{D}_{\text{aggr}}| = 5$ in our main experiments to balance computational efficiency with performance gains.

*Table 11.* Effect of aggregation size.

| Method \ Task | Winowhy | Epistemic_reasoning | AQuA | Trec |
|---|---|---|---|---|
| AUTOSELECT ($|\mathcal{D}_{\text{aggr}}|$: 3) | 0.641 | 0.589 | 0.383 | 0.379 |
| AUTOSELECT ($|\mathcal{D}_{\text{aggr}}|$: 5) | 0.657 | 0.601 | 0.395 | 0.393 |
| AUTOSELECT ($|\mathcal{D}_{\text{aggr}}|$: 7) | 0.664 | 0.595 | 0.403 | 0.404 |
| AUTOSELECT ($|\mathcal{D}_{\text{aggr}}|$: 10) | 0.662 | 0.607 | 0.418 | 0.428 |

**Temperature.** In Table 12, we set the initial scaling parameter $\gamma$ to values in $\{0.02, 0.05, 0.1, 0.2\}$. We follow our original experiment settings by applying a linear scheduler to the scaling parameter $\gamma$. We start from the initial value and increase linearly to $\gamma = 1$ over the first 200 episodes. Here, an initial $\gamma = 0.1$ consistently yields strong performance across tasks. Setting the initial scaling parameter too low (e.g., 0.02) results in worse performance, likely because the distribution becomes too flat (uniform), leading to excessive noise during the early stages of training. Conversely, starting with a higher scaling parameter (0.2) works well for some tasks (AQuA) but degrades performance on others (e.g., Trec), suggesting that $\gamma = 0.1$ provides the best balance between exploration and exploitation.

*Table 12.* Effect of initial temperature.

| Method \ Task | Winowhy | Epistemic_reasoning | AQuA | Trec |
|---|---|---|---|---|
| AUTOSELECT (initial $\gamma$: 0.2) | 0.648 | 0.589 | 0.409 | 0.371 |
| AUTOSELECT (initial $\gamma$: 0.1) | 0.657 | 0.601 | 0.395 | 0.393 |
| AUTOSELECT (initial $\gamma$: 0.05) | 0.647 | 0.577 | 0.398 | 0.411 |
| AUTOSELECT (initial $\gamma$: 0.02) | 0.621 | 0.542 | 0.373 | 0.334 |

**Replay Buffer Size.** From Table 13, we find that a replay buffer size of 50 generally offers the optimal trade-off. Very small buffers (10) can lead to instability due to high correlation in the sampled batches. Meanwhile, increasing the buffer size too much (100) does not monotonically improve performance. One possible reason is that an excessively large buffer retains too many older trajectories generated by earlier and less-optimal policy versions. This dilutes the focus on recent, higher-quality trajectories, thereby hindering the policy from adapting efficiently to the current information.

*Table 13.* Effect of replay buffer size.

| Method \ Task | Winowhy | Epistemic_reasoning | AQuA | Trec |
|---|---|---|---|---|
| AUTOSELECT (Replay Buffer Size: 10) | 0.652 | 0.577 | 0.388 | 0.361 |
| AUTOSELECT (Replay Buffer Size: 30) | 0.636 | 0.563 | 0.372 | 0.385 |
| AUTOSELECT (Replay Buffer Size: 50) | 0.657 | 0.601 | 0.395 | 0.393 |
| AUTOSELECT (Replay Buffer Size: 100) | 0.638 | 0.575 | 0.405 | 0.377 |

## B.3. Efficiency-Performance Analysis of Selection Methods

**Visualization of Demonstration Selection.** Fig. 10 visualizes the average trajectory lengths per dataset. AUTOSELECT adapts to task-specific characteristics by selecting demonstration sequences of varying lengths, where shorter trajectories are generally selected for less difficult tasks (e.g., "SST-2") and longer ones for challenging reasoning tasks (e.g., "Winowhy"). This highlights AUTOSELECT's ability to tailor its selections to the complexity and reasoning demands of each task, while effectively balancing task performance and the computational cost of LLM inference by adaptively adjusting the number of exemplars within the context window.

**Discussion and Future Work (Long-context / Many-shot In-Context Learning).** Recall that our work aims at few-shot demonstration selection, where the goal is to maximize accuracy under a limited demonstration budget, since the few-shot exemplar setting is often treated as the *efficiency sweet spot* for moderate-sized LLMs (Purohit et al., 2025). The computational cost of our auto-regressive approach, which scales with sequence length, is well-aligned with

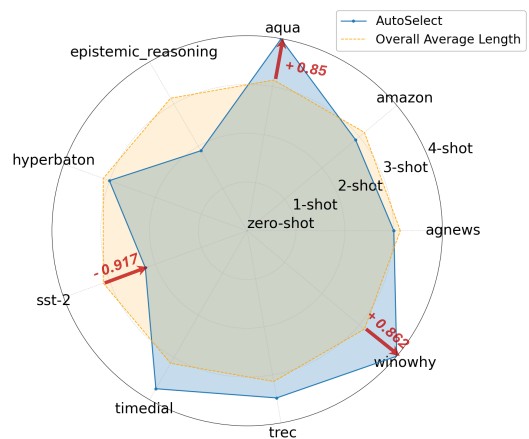

*Figure 10.* Average trajectory length on different datasets given the maximum length $T = 4$.

the targeted *few-shot in-context learning* paradigm where performance relies on moderate-length and high-quality demonstration sequences. This efficiency is further enhanced by an integrated *early stopping mechanism* that dynamically learns to terminate the selection process, often yielding minimal, cost-effective sequences as confirmed by the above analysis.

By contrast, the many-shot regime (Agarwal et al., 2024) operates at a substantially different scale and cost: it requires large context windows, evaluates hundreds to thousands of shots, and often relies on very large token budgets for best performance. Thus, broad many-shot validity is beyond the scope of this paper rather than a claim of our current setting. While efficiently extending this framework to long-context applications such as Retrieval-Augmented Generation (RAG) (Xu et al., 2024) remains challenging due to the potential computational complexity of auto-regressive exemplar selection, we view this as a promising direction for future work. In particular, we plan to investigate architectural modifications. First, we can equip the auto-regressive policy with *sparse or structured attention* mechanisms, such as sparse attention patterns (Child et al., 2019) or block-local and global attention schemes tailored to long documents (Beltagy et al., 2020). This design aims to focus computation on the most relevant portions of the context prefix. Second, we can introduce a *context abstraction layer* that compresses the growing prefix into a compact state representation, inspired by hierarchical memory mechanisms for long-range sequence modeling (He et al., 2025). These extensions could help AUTOSELECT scale to considerably larger values of $T$ in long-context RAG settings; in its current form, however, AUTOSELECT is intentionally designed and optimized for the efficiency requirements of our targeted few-shot ICL setting.

**Inference-time Results.** Meanwhile, we compare AUTOSELECT (using its learned policy to select up to $T = 4$ exemplars for each query) against a random baseline at a larger scale ($T = 16$), as an application-oriented budget comparison. We measure both Test Accuracy and Relative Inference Cost on 400 test samples of AQuA. Table 14 shows that AUTOSELECT achieves a higher accuracy with a lower cost. Our method with a maximum of 4 exemplars outperforms a random baseline that uses 16 exemplars. The above computational advantage can intuitively become more significant as the language model scale increases, since larger models are generally more computationally demanding at inference time.

| Method | Exemplars ($k$-shot) | Test Accuracy | Inference Time Cost |
|---|---|---|---|
| Random | 16 | 0.388 | 1 min 53 seconds ($\sim$**1.76x**) |
| **AUTOSELECT** | **4** | **0.395** | 1 min 27 seconds (**1.0x**) |

*Table 14.* Inference running time and accuracy comparison on 400 test samples of AQuA.

The main "added complexity" of our method lies in its *training phase*, which requires a tiny number of samples from the validation set in each episode to provide feedback signals (rewards) for learning. This is a *standard and necessary practice* in this line of research (Zhang et al., 2022; Wu et al., 2024; Chen et al., 2024b) to guide policy optimization. Thus, AUTOSELECT's one-time policy training is justified by downstream inference savings and improved accuracy.

**Training and inference-time cost.** We analyze both the offline training cost and deployment-time inference cost of AUTOSELECT. At inference time, we compare AUTOSELECT with retrieval-based BM25 under Qwen2.5-3B, including both exemplar selection and downstream LLM inference. BM25 requires $0.212/0.401/0.627/0.935$ seconds per query at $T = 4/10/16/24$, while AUTOSELECT requires $0.282/0.546/0.841/1.162$ seconds per query under the same shot counts. Thus, AUTOSELECT incurs moderate overhead and follows the same monotonic increase with context length. At deployment, AUTOSELECT does not perform reward evaluation; inference only consists of selector-based exemplar choice followed by downstream LLM prediction.

For offline training, 18.1% of the end-to-end runtime is spent on downstream LLM calls, while the remaining 81.9% is selector-side/model-side computation. Within the LLM-call budget, reward evaluation accounts for 51.9%, validation during training for 43.3%, and the final held-out test for 4.8%. This one-time offline cost learns a reusable selector, while deployment retains only modest test-time overhead.

### B.4. Effectiveness of ViT Backbone Choice

To validate the core choice of a Vision Transformer (ViT) backbone with 2D matrix representations (Appendix A.2), we conduct an ablation study against a more conventional alternative. For this, we implement a variant of our AUTOSELECT framework where the ViT backbone is replaced with a pre-trained *GPT-2 Medium model*, and the 2D matrix embeddings are replaced with standard *flattened 1D token vectors*.

Crucially, this GPT-2 variant is not a simple zero-shot selector; it is also trained using the exact same auto-regressive paradigm and learning procedure as our proposed framework (as outlined in Alg. 1). This ensures that the only differences are the backbone model and the input representation, allowing for a direct and fair comparison of these architectural components.

As shown in Fig. 11, our ViT-based model consistently outperforms the GPT-2 variant, despite the latter having significantly more parameters. This result strongly validates our design, indicating that the synergy between the ViT architecture and 2D matrix representations is more effective at capturing the necessary structural information for this task than a larger, general-purpose transformer operating on flattened data. Notably, both AUTOSELECT variants considerably outperform weaker baselines like ActRL and random selection, underscoring the general effectiveness of our auto-regressive paradigm.

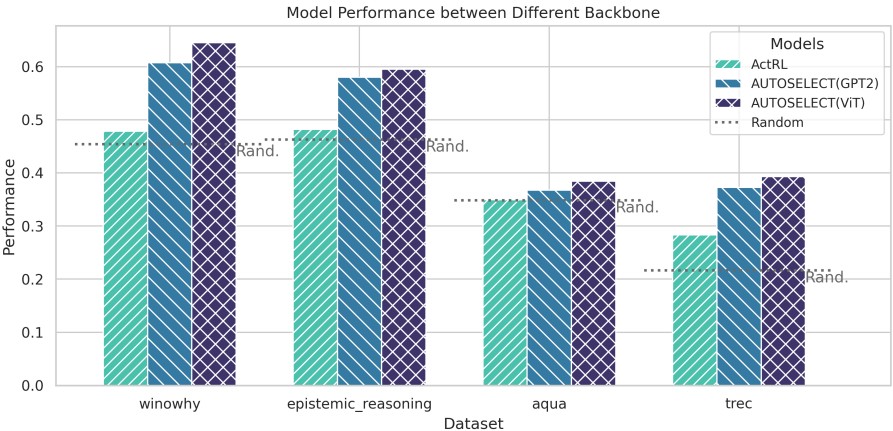

*Figure 11.* Performance evaluation of GPT-2 Medium as an alternative policy backbone. Horizontal line "Rand." (Random) denotes reference performance from Table 1.

**Visualization and Discussion.** We also visualize the learned attention over 2D query-exemplar matrices on the AQuA dataset. In Fig. 12, the first panel is one query matrix and the remaining four panels are the selected exemplars. Each heatmap is annotated with its attention weight ("w" value) and its row/column correlations ($r_{row}$, $r_{col}$) with the query. We observe that the first three exemplars show strong structural alignment with the query (high $r_{row}$ / $r_{col}$ and high weight). This means that the ViT is explicitly matching 2D patterns across both axes, capturing how the "question-reasoning-answer" structure maps onto the exemplars, rather than just comparing token-by-token. By contrast, the fourth exemplar, which is selected last in the auto-regressive trajectory, has a clearly mismatched 2D structure, low (even negative) correlations, and a much smaller attention weight. This indicates that although it is the argmax under the model's scoring at that step, its

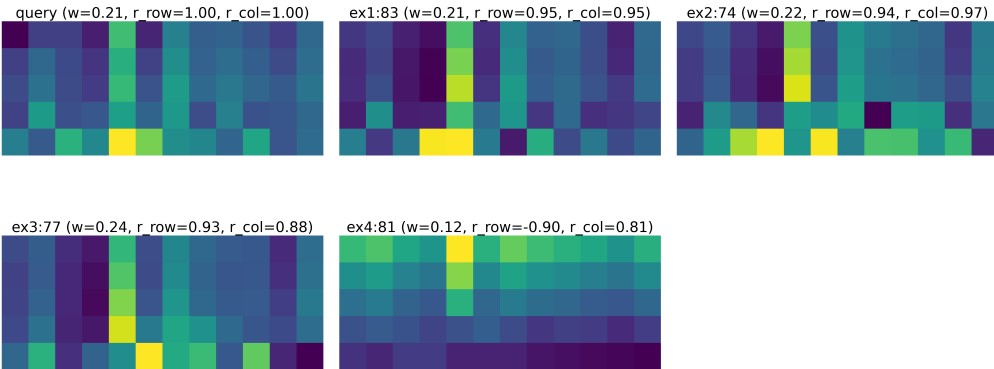

*Figure 12.* Visualization of AUTOSELECT's 2D attention on the AQuA dataset: matrices (first panel: one query; remaining panels: four selected exemplars with attention weights and row/column correlations to the query). AUTOSELECT assigns higher weight ("w" value) to structurally aligned exemplars while down-weighting a structurally mismatched and negatively correlated last exemplar, which highlights the advantage of operating on 2D matrices.

contribution is effectively down-weighted when aggregating information. This also *suggests that an early-termination (EOS) signal can be beneficial to stop before adding such marginal exemplars*. This kind of spatially coherent and structure-aware matching is hard to realize with a GPT-2-style policy over flattened vectors, where the 2D layout is destroyed and long-range dependencies need to be inferred over a single 1D sequence. Together with the ablation results with the GPT-2 architecture (Fig. 11), these attention maps provide evidence that the ViT policy can effectively leverage the matrix structure to model inter-exemplar relationships.

### B.5. Examples of Selected Exemplar Sequences: Correlations between Selected Exemplars and Input Queries

In this subsection, we present several examples of input queries along with their corresponding exemplar sequences selected by AUTOSELECT, as shown in Tables 15–18 below. We also provide insights into the possible rationale behind the exemplar selection outcome.

| | |
|---|---|
| **Input Query** | The ratio of the present ages of a son and his father is 1 : 5 and that of his mother and father is 4 : 5. After 2 years the ratio of the age of the son to that of his mother becomes 3 : 10. What is the present age of the father? |
| **Exemplar 1** | The credit card and global payment processing companies have been suffering losses for some time now. A well-known company recently announced its quarterly results. According to the results, the revenue fell to $48.0 billion from $69.0 billion, a year ago. By what percent did the revenue fall? |
| **Exemplar 2** | Four friends, Peter, John, Quincy, and Andrew, are pooling their money to buy a $1600 item. Peter has twice as much money as John. Quincy has $40 more than Peter. Andrew has 10% more than Quincy. If they put all their money together and spend the $1600, they will have $14 left. How much money does Peter have? |
| **Exemplar 3** | Lagaan is levied on the 60 percent of the cultivated land. The revenue department collected total Rs. 3,74,000 through the lagaan from the village of Mutter. Mutter, a very rich farmer, paid only Rs. 480 as lagaan. The percentage of total land of Mutter over the total taxable land of the village is: |
| **Exemplar 4** | An exam consists of 8 true/false questions. Brian forgets to study, so he must guess blindly on each question. If any score above 60% is a passing grade, what is the probability that Brian passes? |

*Table 15:* Demonstration of one exemplary input query and corresponding four chosen exemplars for AQuA dataset. These exemplars are particularly effective because they all involve percentage or ratio information directly correlating with the input query, while simultaneously presenting diverse problem-solving structures. This combination of relevant numerical concepts across various contexts also reinforces the multi-step relational reasoning needed for the input query.

| | |
|---|---|
| **Input Query** | What is the capital of Yugoslavia? |
| **Exemplar 1** | What is the address for the main government office in Rome, Italy? |
| **Exemplar 2** | What happened to Pompeii? |
| **Exemplar 3** | How many zip codes are there in the U.S.? |
| **Exemplar 4** | Where did Indian Pudding come from? |

*Table 16:* Demonstration of one exemplary input query and corresponding four chosen exemplars for Trec dataset. This exemplar set is particularly beneficial for the query "What is the capital of Yugoslavia?", because the chosen exemplars directly align with the query's need for a specific location-based factual answer, sharing the same broad category of "location". Meanwhile, the remaining exemplars seek different types of information, which helps clarify the query's intent by highlighting its focus on retrieving a geographical entity. This makes the chosen exemplar sequence more instructive than a random or less targeted selection.

| | |
|---|---|
| **Input Query** | The delivery truck zoomed by the school bus because it was going so fast. The 'it' refers to the delivery truck because the delivery truck was faster. |
| **Exemplar 1** | The table won't fit through the doorway because it is too narrow. The 'it' refers to the doorway because the doorway won't fit through the table because it is too wide. |
| **Exemplar 2** | The dog chased the cat, which ran up a tree. It waited at the top. The 'It' refers to the cat because It waited at the top. The dog chased the cat, which ran up a tree. |
| **Exemplar 3** | Tom said "Check" to Ralph as he took his bishop. The 'his' refers to Ralph because that's the only way we can understand it in the game. |
| **Exemplar 4** | Fred was supposed to run the dishwasher, but he put it off, because he wanted to watch TV. But the show turned out to be boring, so he changed his mind and turned it on. The 'it' refers to the dishwasher because the TV was already on. The dishwasher wasn't. So he chose to turn the dishwasher on. |

*Table 17:* Demonstration of one exemplary input query and corresponding four chosen exemplars for Winowhy dataset. This exemplar set is particularly insightful, because it reflects the input query's core reliance on comparative reasoning, such as a truck being "faster" or a table being "too wide" versus a doorway being "too narrow" in Exemplar 1. This helps determine pronoun reference based on contrasting attributes or actions. The diverse comparison contexts across the exemplars, along with Winowhy's characteristic patterns (as seen in Exemplars 1, 3, and 4), help the model resolve the pronoun in the query using similar comparative logic. They also guide the model to produce reasoning that aligns with the dataset's common patterns.

| Input Query | Premise: Charles assumes that Isabella sees that a man is resting in a small stream with a hat over his head while the little waterfall is pouring in the background. Hypothesis: Charles assumes that a man is resting in a small stream with a hat over his head while the little waterfall is pouring in the background. |
|---|---|
| Exemplar 1 | Premise: John thinks that Isabella knows that boy in green pajamas plays with his toy while his mother sits on the couch and watches him. Hypothesis: John thinks that boy in green pajamas plays with his toy while his mother sits on the couch and watches him. |
| Exemplar 2 | Premise: John sees that a basketball player in white and red goes up for the shot as two defensive men in red jump up to block the shot. Hypothesis: John sees that a basketball player is trying to make a shot while two players try to block it. |
| Exemplar 3 | Premise: Charles believes that Evelyn learns that a person dressed in winter clothes poses with a snowman surrounded by snow covered landscape. Hypothesis: Evelyn learns that a person dressed in winter clothes poses with a snowman surrounded by snow covered landscape. |
| Exemplar 4 | Premise: Taylor suspects that Olivia thinks that a man on one miniature train passes a circular object to a man on another train as they pass by. Hypothesis: Taylor suspects that a man on one miniature train passes a circular object to a man on another train as they pass by. |

*Table 18:* Demonstration of one exemplary input query and corresponding four chosen exemplars for Epistemic_reasoning dataset. The above example illustrates how contrasting exemplars with similar structures can support epistemic understanding in language models. For instance, one exemplar ("John thinks Isabella knows...") mirrors the query's logic and provides a useful reference. Meanwhile, another exemplar ("Taylor suspects Olivia...") applies a similar structural transformation but leads to an incorrect inference. This contrast, paired with an additional exemplar illustrating a different type of invalid simplification, provides valuable guidance for the LLM's inference process.

## C. Equivalence of Policy-level Cross-Entropy (CE) Loss Minimization and Induced Plackett-Luce (PL) Ranking Optimization

In this section, we provide a detailed description and proof of Theorem 4.2 from the main body, which establishes the connection between minimizing the cross-entropy loss and optimizing the Plackett-Luce (PL) ranking, conditioned on a collected trajectory set $\mathcal{T}$. Subsequently, we present the following Theorem C.1, which corresponds to Theorem 4.2 in the main body.

---

**Theorem C.1.** *Suppose $\pi^*(\cdot|\boldsymbol{x})$ and $\pi_\theta(\cdot|\boldsymbol{x})$ are two probability distributions induced by two policies $\pi^*, \pi_\theta$ respectively. Let $\mathcal{T}$ be a non-empty finite collection of trajectories such that the total probability masses $Z^* = \sum_{\tau \in \mathcal{T}} \pi^*(\tau|\boldsymbol{x})$ and $Z_\theta = \sum_{\tau \in \mathcal{T}} \pi_\theta(\tau|\boldsymbol{x})$ are strictly positive. Define the conditional distributions over $\mathcal{T}$ as $\pi^*(\tau|\mathcal{T}, \boldsymbol{x}) = \pi^*(\tau|\boldsymbol{x})/Z^*$ and $\pi_\theta(\tau|\mathcal{T}, \boldsymbol{x}) = \pi_\theta(\tau|\boldsymbol{x})/Z_\theta, \forall \tau \in \mathcal{T}$. The probability factor is denoted by $\epsilon := \min\{\pi^*(\tau_{\max}|\mathcal{T}, \boldsymbol{x}), \pi_\theta(\tau_{\max}|\mathcal{T}, \boldsymbol{x})\}$ where $\tau_{\max} = \arg\max_{\tau' \in \mathcal{T}} |\log \frac{\pi^*(\tau'|\mathcal{T}, \boldsymbol{x})}{\pi_\theta(\tau'|\mathcal{T}, \boldsymbol{x})}|$. Then, the absolute difference between PL ranking models can be bounded by*

$$\max_\sigma \left| P_{\pi^*}(\sigma \mid \mathcal{T}, \boldsymbol{x}) - P_{\pi_\theta}(\sigma \mid \mathcal{T}, \boldsymbol{x}) \right| \le \frac{2\beta \cdot |\mathcal{T}|}{\epsilon} \cdot \sqrt{2 \cdot (\mathcal{L}_{CE}^{\mathcal{T}}(\pi^*, \pi_\theta) - \mathcal{L}_{CE}^{\mathcal{T}}(\pi^*, \pi^*))}, \tag{13}$$

*where $\mathcal{L}_{CE}^{\mathcal{T}}(\pi_1, \pi_2)$ refers to the cross-entropy (CE) loss given target policy $\pi_1$ and trainable policy $\pi_2$ over trajectory collection $\mathcal{T}$.*

---

*Proof.* To begin with, let $f(p) = \log(p)$ for $p > 0$. By the Mean Value Theorem, if $f$ is continuous on $[a, b]$ and differentiable on $(a, b)$, then there exists $c \in (a, b)$ such that

$$f(b) - f(a) = f'(c)(b - a) \tag{14}$$

Using Lemma C.2 while denoting $a = \pi_\theta(\tau|\mathcal{T}, \boldsymbol{x})$ and $b = \pi^*(\tau|\mathcal{T}, \boldsymbol{x})$, we have

$$\log(\pi^*(\tau|\mathcal{T}, \boldsymbol{x})) - \log(\pi_\theta(\tau|\mathcal{T}, \boldsymbol{x})) = \frac{1}{c}(\pi^*(\tau|\mathcal{T}, \boldsymbol{x}) - \pi_\theta(\tau|\mathcal{T}, \boldsymbol{x}))$$

$$\log \frac{\pi^*(\tau|\mathcal{T}, \boldsymbol{x})}{\pi_\theta(\tau|\mathcal{T}, \boldsymbol{x})} = \frac{\pi^*(\tau|\mathcal{T}, \boldsymbol{x}) - \pi_\theta(\tau|\mathcal{T}, \boldsymbol{x})}{c}$$

Since $c$ is between $\pi_\theta(\tau|\mathcal{T}, \boldsymbol{x})$ and $\pi^*(\tau|\mathcal{T}, \boldsymbol{x})$, we know $c \ge \min\{\pi_\theta(\tau|\mathcal{T}, \boldsymbol{x}), \pi^*(\tau|\mathcal{T}, \boldsymbol{x})\}$. As $\frac{1}{c}$ decreases with increasing $c$, we have

$$\left| \log \frac{\pi^*(\tau|\mathcal{T}, \boldsymbol{x})}{\pi_\theta(\tau|\mathcal{T}, \boldsymbol{x})} \right| = \left| \frac{\pi^*(\tau|\mathcal{T}, \boldsymbol{x}) - \pi_\theta(\tau|\mathcal{T}, \boldsymbol{x})}{c} \right|$$

$$\le \frac{|\pi^*(\tau|\mathcal{T}, \boldsymbol{x}) - \pi_\theta(\tau|\mathcal{T}, \boldsymbol{x})|}{\min\{\pi^*(\tau|\mathcal{T}, \boldsymbol{x}), \pi_\theta(\tau|\mathcal{T}, \boldsymbol{x})\}}.$$

Next, by applying Pinsker's inequality, for any distributions $p$ and $q$, we have $\max_x |p(x) - q(x)| \le \sqrt{2 \cdot D_{KL}(p\|q)}$. In this context, for policies conditioned on trajectories $\mathcal{T}$, denote $\epsilon := \min\{\pi^*(\tau_{\max}|\mathcal{T}, \boldsymbol{x}), \pi_\theta(\tau_{\max}|\mathcal{T}, \boldsymbol{x})\}$ where $\tau_{\max} = \arg\max_{\tau' \in \mathcal{T}} \left| \log \frac{\pi^*(\tau'|\mathcal{T}, \boldsymbol{x})}{\pi_\theta(\tau'|\mathcal{T}, \boldsymbol{x})} \right|$. Then, with the conditional distributions over $\mathcal{T}$ as $P(\tau) := \pi^*(\tau|\mathcal{T}, \boldsymbol{x})$ and $Q(\tau) := \pi_\theta(\tau|\mathcal{T}, \boldsymbol{x})$, we have

$$\max_{\tau \in \mathcal{T}} \left| \log \frac{\pi^*(\tau|\mathcal{T}, \boldsymbol{x})}{\pi_\theta(\tau|\mathcal{T}, \boldsymbol{x})} \right| \le \frac{\sqrt{2 D_{\mathrm{KL}}(P \| Q)}}{\epsilon}. \tag{15}$$

Combining the above derivation with the property $D_{\mathrm{KL}}(P \| Q) = \mathcal{L}_{\mathrm{CE}}^{\mathcal{T}}(\pi^*, \pi_\theta) - \mathcal{L}_{\mathrm{CE}}^{\mathcal{T}}(\pi^*, \pi^*)$, together with the conclusion from Lemma C.2, completes the proof.

$\square$

**Discussion.** The above theorem suggests that minimizing the CE loss over the generated trajectory collection $\mathcal{T}$ can serve as a feasible training objective for optimizing PL rankings. Furthermore, we can consider that the concentrability condition holds for trajectories and policy models, where analogous conditions are commonly adopted in existing reinforcement learning analyses (e.g., (Chen & Jiang, 2019; Hong et al., 2024)), in order to facilitate theoretical analysis and ensure the stability of policy optimization. Here, if we have $\|\pi^*(\tau|\mathcal{T}, \boldsymbol{x})/\pi_\theta(\tau|\mathcal{T}, \boldsymbol{x})\|_\infty \leq \epsilon'$ over trajectories $\tau \in \mathcal{T}$, we can further lower-bound this factor by defining $\epsilon := \min\{\pi^*(\tau_{\max}|\mathcal{T}, \boldsymbol{x}), \pi^*(\tau_{\max}|\mathcal{T}, \boldsymbol{x})/\epsilon'\}$, which subsequently makes the probability factor $\epsilon$ independent of $\pi_\theta$. This consequently makes the upper bound on the RHS of Eq. 13 decrease monotonically as the excess CE loss decreases toward zero, while also making the upper bound independent of the policy $\pi_\theta$. Note that this concentrability condition is also realizable in practice, for example, by incorporating exploration techniques such as epsilon-greedy (Dann et al., 2022) into the trajectory generation process.

---

**Lemma C.2.** *Let $\pi^*(\cdot|\boldsymbol{x})$ and $\pi_\theta(\cdot|\boldsymbol{x})$ be two policy probability distributions. Let $\mathcal{T}$ be a non-empty finite subset such that the total probability masses $Z^* = \sum_{\tau \in \mathcal{T}} \pi^*(\tau|\boldsymbol{x})$ and $Z_\theta = \sum_{\tau \in \mathcal{T}} \pi_\theta(\tau|\boldsymbol{x})$ are strictly positive. Define the conditional distributions over $\mathcal{T}$ as $\pi^*(\tau|\mathcal{T}, \boldsymbol{x}) = \pi^*(\tau|\boldsymbol{x})/Z^*$ and $\pi_\theta(\tau|\mathcal{T}, \boldsymbol{x}) = \pi_\theta(\tau|\boldsymbol{x})/Z_\theta, \forall \tau \in \mathcal{T}$. For any permutation $\sigma$ of the trajectory collection $\mathcal{T}$, the absolute difference, between the probabilities assigned by the Plackett-Luce (PL) ranking models induced by policies $\pi^*$ and $\pi_\theta$, can be bounded by*

$$\left|P_{\pi^*}(\sigma|\, \mathcal{T}, \boldsymbol{x}) - P_{\pi_\theta}(\sigma|\, \mathcal{T}, \boldsymbol{x})\right| \leq 2\beta \cdot |\mathcal{T}| \cdot \max_{\tau' \in \{\tau_1, \ldots, \tau_{|\mathcal{T}|}\}} \left|\log \frac{\pi^*(\tau'|\mathcal{T}, \boldsymbol{x})}{\pi_\theta(\tau'|\mathcal{T}, \boldsymbol{x})}\right|. \tag{16}$$

*where $|\mathcal{T}|$ is the cardinality of trajectory collection $\mathcal{T}$.*

---

*Proof.* Using the definition of the Plackett-Luce (PL) model induced by a policy model $\pi$ and the shift-invariance property of the softmax function, we have

$$
\begin{aligned}
P_\pi(\sigma|\, \mathcal{T}, \boldsymbol{x}) &= \prod_{i=1}^{|\mathcal{T}|} \frac{\exp\left(\beta \log \frac{\pi(\tau_{\sigma(i)}|\boldsymbol{x})}{\pi_{\mathrm{old}}(\tau_{\sigma(i)}|\boldsymbol{x})}\right)}{\sum_{j=i}^{|\mathcal{T}|} \exp\left(\beta \log \frac{\pi(\tau_{\sigma(j)}|\boldsymbol{x})}{\pi_{\mathrm{old}}(\tau_{\sigma(j)}|\boldsymbol{x})}\right)} \\
&= \prod_{i=1}^{|\mathcal{T}|} \frac{\exp\left(\beta \log \frac{\pi(\tau_{\sigma(i)}|\boldsymbol{x})/\sum_{\tau' \in \mathcal{T}} \pi(\tau'|\boldsymbol{x})}{\pi_{\mathrm{old}}(\tau_{\sigma(i)}|\boldsymbol{x})}\right)}{\sum_{j=i}^{|\mathcal{T}|} \exp\left(\beta \log \frac{\pi(\tau_{\sigma(j)}|\boldsymbol{x})/\sum_{\tau' \in \mathcal{T}} \pi(\tau'|\boldsymbol{x})}{\pi_{\mathrm{old}}(\tau_{\sigma(j)}|\boldsymbol{x})}\right)} = \prod_{i=1}^{|\mathcal{T}|} \frac{\exp\left(\beta \log \frac{\pi(\tau_{\sigma(i)}|\mathcal{T}, \boldsymbol{x})}{\pi_{\mathrm{old}}(\tau_{\sigma(i)}|\boldsymbol{x})}\right)}{\sum_{j=i}^{|\mathcal{T}|} \exp\left(\beta \log \frac{\pi(\tau_{\sigma(j)}|\mathcal{T}, \boldsymbol{x})}{\pi_{\mathrm{old}}(\tau_{\sigma(j)}|\boldsymbol{x})}\right)},
\end{aligned}
\tag{17}
$$

and a similar procedure can also be applied to the denominator term to obtain $\pi_{\mathrm{old}}(\tau_{\sigma(j)}|\mathcal{T}, \boldsymbol{x})$. Next, let us denote $s_i(\pi) = \exp\left(\beta \log \frac{\pi(\tau_{\sigma(i)}|\mathcal{T}, \boldsymbol{x})}{\pi_{\mathrm{old}}(\tau_{\sigma(i)}|\mathcal{T}, \boldsymbol{x})}\right) = \left(\frac{\pi(\tau_{\sigma(i)}|\mathcal{T}, \boldsymbol{x})}{\pi_{\mathrm{old}}(\tau_{\sigma(i)}|\mathcal{T}, \boldsymbol{x})}\right)^\beta$, and also denote $Z_i(\pi) = \sum_{j=i}^{|\mathcal{T}|} s_j(\pi)$. Then, the PL model probability can be written as

$$P_\pi(\sigma|\, \mathcal{T}, \boldsymbol{x}) = \prod_{i=1}^{|\mathcal{T}|} \frac{s_i(\pi)}{Z_i(\pi)} \tag{18}$$

**Decomposition of the objective.** We then focus on how the probability differs between the optimal policy $\pi^*$ and our learnable policy $\pi_\theta$. Given four positive values $a, b, c, d > 0$, we begin by writing

$$\frac{a}{b} - \frac{c}{d} = \frac{ad - bc}{bd}.$$

One way to split the numerator is to write $ad - bc = d(a - c) + c(d - b)$. Taking absolute values and applying the triangle inequality yields

$$\left|\frac{a}{b} - \frac{c}{d}\right| \leq \frac{d\,|a - c|}{bd} + \frac{c\,|d - b|}{bd} = \frac{|a - c|}{b} + \frac{c}{bd}\,|b - d|.$$

Alternatively, we can write $ad - bc = a(d - b) + b(a - c)$, and we can similarly obtain

$$\left|\frac{a}{b} - \frac{c}{d}\right| \leq \frac{|a - c|}{d} + \frac{a}{bd}\,|b - d|.$$

Taking the minimum of these two bounds gives the final inequality:

$$\left| \frac{a}{b} - \frac{c}{d} \right| \leq \min \left\{ \frac{|a-c|}{b} + \frac{c}{bd} |b-d|, \quad \frac{|a-c|}{d} + \frac{a}{bd} |b-d| \right\}.$$

The above derivation shows that by splitting the numerator in two different ways and applying the triangle inequality, we can obtain two valid bounds, and taking their minimum provides the upper bound.

Subsequently, we can use the above inequality to bound the difference in each factor of the product. In particular, for each $i \in \{1, \ldots, |\mathcal{T}|\}$, we can formulate the bound as

$$\left| \frac{s_i(\pi^*)}{Z_i(\pi^*)} - \frac{s_i(\pi_\theta)}{Z_i(\pi_\theta)} \right| \leq \begin{cases} \dfrac{|s_i(\pi^*) - s_i(\pi_\theta)|}{Z_i(\pi^*)} + \dfrac{s_i(\pi_\theta)}{Z_i(\pi^*)\,Z_i(\pi_\theta)} |Z_i(\pi^*) - Z_i(\pi_\theta)|, & \text{if } s_i(\pi^*) \geq s_i(\pi_\theta), \\[2ex] \dfrac{|s_i(\pi^*) - s_i(\pi_\theta)|}{Z_i(\pi_\theta)} + \dfrac{s_i(\pi^*)}{Z_i(\pi^*)\,Z_i(\pi_\theta)} |Z_i(\pi^*) - Z_i(\pi_\theta)|, & \text{if } s_i(\pi^*) < s_i(\pi_\theta). \end{cases} \tag{19}$$

Next, without loss of generality, we consider $s_i(\pi^*) \geq s_i(\pi_\theta)$ for the proof below while applying the first inequality in Eq. 19. We also note the other case $s_i(\pi^*) < s_i(\pi_\theta)$ can be readily proved by following an analogous procedure, by alternatively applying the second inequality in Eq. 19. We then proceed to bound $|s_i(\pi^*) - s_i(\pi_\theta)|$ and $|Z_i(\pi^*) - Z_i(\pi_\theta)|$.

To begin with, **(1) for the first term**, we set $a = \frac{\pi^*(\tau_{\sigma(i)}|\mathcal{T},\boldsymbol{x})}{\pi_{\text{old}}(\tau_{\sigma(i)}|\mathcal{T},\boldsymbol{x})}$, $b = \frac{\pi_\theta(\tau_{\sigma(i)}|\mathcal{T},\boldsymbol{x})}{\pi_{\text{old}}(\tau_{\sigma(i)}|\mathcal{T},\boldsymbol{x})}$, so that $s_i(\pi^*) = a^\beta$, and $s_i(\pi_\theta) = b^\beta$. By the Mean Value Theorem on $f(x) = x^\beta$, there exists some $\xi$ strictly between $a$ and $b$ such that $a^\beta - b^\beta = \beta \xi^{\beta-1}(a - b)$. Since $\beta > 0$ and $\xi > 0$ (as it is between $a, b > 0$), $\xi^{\beta-1} > 0$. Thus, taking absolute values gives:

$$|a^\beta - b^\beta| = \beta \xi^{\beta-1} |a - b|. \quad (*)$$

Next, applying the Mean Value Theorem to $g(x) = \log x$, there exists some $\eta$ strictly between $a$ and $b$ such that $\log a - \log b = \frac{1}{\eta}(a - b)$. Since $\eta > 0$:

$$|a - b| = \eta |\log a - \log b|. \quad (**)$$

Substituting equation $(**)$ into equation $(*)$ yields:

$$|a^\beta - b^\beta| = \beta(\xi^{\beta-1}\eta)|\log a - \log b|.$$

To evaluate the term $\xi^{\beta-1}\eta$, we then apply Cauchy's Mean Value Theorem, such that for $F(x) = x^\beta$ and $G(x) = \log x$, there exists $c$ strictly between $a$ and $b$ such that

$$\frac{F(a) - F(b)}{G(a) - G(b)} = \frac{F'(c)}{G'(c)} \implies \frac{a^\beta - b^\beta}{\log a - \log b} = \frac{\beta c^{\beta-1}}{1/c} = \beta c^\beta.$$

Thus, $\frac{|a^\beta - b^\beta|}{|\log a - \log b|} = \beta c^\beta$ (since $\beta > 0, c^\beta > 0$). Comparing this with our combined expression, we have $\xi^{\beta-1}\eta = c^\beta$. Since $c$ is strictly between $a$ and $b$, and the function $h(x) = x^\beta$ is strictly increasing for $x > 0$ (because $\beta > 0$), it follows that $c^\beta < \max\{a^\beta, b^\beta\}$. Therefore, $c^\beta \leq \max\{a^\beta, b^\beta\}$. Substituting $\xi^{\beta-1}\eta = c^\beta$ and applying the above results, we consequently have

$$|a^\beta - b^\beta| = \beta c^\beta |\log a - \log b| \leq \beta \max\{a^\beta, b^\beta\}|\log a - \log b|.$$

Returning to our original notation, we therefore obtain

$$|s_i(\pi^*) - s_i(\pi_\theta)| \leq \beta \max\{s_i(\pi^*), \, s_i(\pi_\theta)\} \left| \log \frac{\pi^*(\tau_{\sigma(i)} \mid \mathcal{T}, \boldsymbol{x})}{\pi_\theta(\tau_{\sigma(i)} \mid \mathcal{T}, \boldsymbol{x})} \right|.$$

**(2) Similarly, for the second term**, we have

$$|Z_i(\pi^*) - Z_i(\pi_\theta)| = \left| \sum_{j=i}^{|\mathcal{T}|} (s_j(\pi^*) - s_j(\pi_\theta)) \right|$$

$$\leq \sum_{j=i}^{|\mathcal{T}|} |s_j(\pi^*) - s_j(\pi_\theta)| \leq \beta \cdot \sum_{j=i}^{|\mathcal{T}|} \max\{s_j(\pi^*), s_j(\pi_\theta)\} \cdot \left| \log \frac{\pi^*(\tau_{\sigma(j)}|\mathcal{T}, \boldsymbol{x})}{\pi_\theta(\tau_{\sigma(j)}|\mathcal{T}, \boldsymbol{x})} \right|$$

Let $\delta = \max_{\tau' \in \{\tau_1, \dots, \tau_{|\mathcal{T}|}\}} \left| \log \frac{\pi^*(\tau' | \mathcal{T}, \boldsymbol{x})}{\pi_\theta(\tau' | \mathcal{T}, \boldsymbol{x})} \right|$, which is related to the maximum log-ratio between the optimal and trainable policies across collected trajectories. Then

$$|s_i(\pi^*) - s_i(\pi_\theta)| \le \beta \cdot \max\{s_i(\pi^*), s_i(\pi_\theta)\} \cdot \delta$$

$$|Z_i(\pi^*) - Z_i(\pi_\theta)| \le \beta \cdot \sum_{j=i}^{|\mathcal{T}|} \max\{s_j(\pi^*), s_j(\pi_\theta)\} \cdot \delta$$

Substituting these bounds back gives

$$\left| \frac{s_i(\pi^*)}{Z_i(\pi^*)} - \frac{s_i(\pi_\theta)}{Z_i(\pi_\theta)} \right| \le \frac{\beta \cdot \max\{s_i(\pi^*), s_i(\pi_\theta)\} \cdot \delta}{Z_i(\pi^*)}$$

$$+ \frac{s_i(\pi_\theta)}{Z_i(\pi^*)Z_i(\pi_\theta)} \cdot \beta \cdot \sum_{j=i}^{|\mathcal{T}|} \max\{s_j(\pi^*), s_j(\pi_\theta)\} \cdot \delta$$

We can further simplify this bound to

$$\left| \frac{s_i(\pi^*)}{Z_i(\pi^*)} - \frac{s_i(\pi_\theta)}{Z_i(\pi_\theta)} \right| \le \beta \cdot \delta \cdot \left( \frac{\max\{s_i(\pi^*), s_i(\pi_\theta)\}}{Z_i(\pi^*)} + \frac{s_i(\pi_\theta) \cdot \sum_{j=i}^{|\mathcal{T}|} \max\{s_j(\pi^*), s_j(\pi_\theta)\}}{Z_i(\pi^*)Z_i(\pi_\theta)} \right)$$

Here, since we consider $s_i(\pi^*) \ge s_i(\pi_\theta)$ without loss of generality for the proof, we have

$$\frac{\max\{s_i(\pi^*), s_i(\pi_\theta)\}}{Z_i(\pi^*)} = \frac{s_i(\pi^*)}{Z_i(\pi^*)} \le 1.$$

Meanwhile, for the second term, we have

$$\frac{s_i(\pi_\theta) \cdot \sum_{j=i}^{|\mathcal{T}|} \max\{s_j(\pi^*), s_j(\pi_\theta)\}}{Z_i(\pi^*)Z_i(\pi_\theta)} = \frac{s_i(\pi_\theta)s_i(\pi^*) + s_i(\pi_\theta) \cdot \sum_{j=i+1}^{|\mathcal{T}|} \max\{s_j(\pi^*), s_j(\pi_\theta)\}}{Z_i(\pi^*)Z_i(\pi_\theta)} \le 1,$$

where in this expression, each product term in the numerator appears in the decomposition of the denominator (by appropriately relaxing $s_i(\pi_\theta)$ to $s_i(\pi^*)$ when needed). Thus, we also have $\frac{s_i(\pi_\theta) \cdot \sum_{j=i}^{|\mathcal{T}|} \max\{s_j(\pi^*), s_j(\pi_\theta)\}}{Z_i(\pi^*)Z_i(\pi_\theta)} \le 1$.

As we have mentioned previously, the above results can also be derived analogously when $s_i(\pi^*) < s_i(\pi_\theta)$, by alternatively adopting the second inequality in Eq. 19. As a result, we have

$$\left| \frac{s_i(\pi^*)}{Z_i(\pi^*)} - \frac{s_i(\pi_\theta)}{Z_i(\pi_\theta)} \right| \le 2\beta \cdot \delta.$$

Then, recall the full product is given by

$$|P_{\pi^*}(\sigma | \mathcal{T}, \boldsymbol{x}) - P_{\pi_\theta}(\sigma | \mathcal{T}, \boldsymbol{x})| = \left| \prod_{i=1}^{|\mathcal{T}|} \frac{s_i(\pi^*)}{Z_i(\pi^*)} - \prod_{i=1}^{|\mathcal{T}|} \frac{s_i(\pi_\theta)}{Z_i(\pi_\theta)} \right|.$$

We first observe that

$$\prod_{i=1}^{n} A_i - \prod_{i=1}^{n} B_i = \sum_{i=1}^{n} \left( \prod_{j=1}^{i-1} A_j \right) (A_i - B_i) \left( \prod_{j=i+1}^{n} B_j \right).$$

Taking absolute values and applying the triangle inequality gives $\left| \prod_{i=1}^{n} A_i - \prod_{i=1}^{n} B_i \right| \le \sum_{i=1}^{n} |A_i - B_i| \left| \prod_{j=1}^{i-1} A_j \right| \left| \prod_{j=i+1}^{n} B_j \right|$. For each index $j$, we have $A_j, B_j \le \max\{A_j, B_j\}$, and it follows that $\left| \prod_{j=1}^{i-1} A_j \right| \left| \prod_{j=i+1}^{n} B_j \right| \le \prod_{j=1, j \ne i}^{n} \max\{A_j, B_j\}$. In this context, we can obtain the following inequality

$$\left| \prod_{i=1}^{n} A_i - \prod_{i=1}^{n} B_i \right| \le \sum_{i=1}^{n} |A_i - B_i| \prod_{j=1, j \ne i}^{n} \max\{A_j, B_j\}.$$

In our settings, denoting $A_i = \frac{s_i(\pi^*)}{Z_i(\pi^*)}$ and $B_i = \frac{s_i(\pi_\theta)}{Z_i(\pi_\theta)}$, the above derived telescoping-product inequality will consequently lead to

$$
\begin{aligned}
\left| P_{\pi^*}(\sigma \,|\, \mathcal{T}, \boldsymbol{x}) - P_{\pi_\theta}(\sigma \,|\, \mathcal{T}, \boldsymbol{x}) \right| &\leq \sum_{i=1}^{|\mathcal{T}|} \left| \frac{s_i(\pi^*)}{Z_i(\pi^*)} - \frac{s_i(\pi_\theta)}{Z_i(\pi_\theta)} \right| \cdot \prod_{j \neq i} \max \left\{ \frac{s_j(\pi^*)}{Z_j(\pi^*)}, \frac{s_j(\pi_\theta)}{Z_j(\pi_\theta)} \right\} \\
&\leq \sum_{i=1}^{|\mathcal{T}|} 2\beta \cdot \delta \cdot \prod_{j \neq i} 1 \\
&= 2\beta \cdot |\mathcal{T}| \cdot \delta
\end{aligned}
$$

where the first inequality is because each fraction $\frac{s_j(\pi)}{Z_j(\pi)} \leq 1$, and their product is also at most 1. Summarizing all the results gives

$$
\left| P_{\pi^*}(\sigma \,|\, \mathcal{T}, \boldsymbol{x}) - P_{\pi_\theta}(\sigma \,|\, \mathcal{T}, \boldsymbol{x}) \right| \leq 2\beta \cdot |\mathcal{T}| \cdot \max_{\tau' \in \{\tau_1, \ldots, \tau_{|\mathcal{T}|}\}} \left| \log \frac{\pi^*(\tau' | \mathcal{T}, \boldsymbol{x})}{\pi_\theta(\tau' | \mathcal{T}, \boldsymbol{x})} \right|,
$$

which completes the proof.

$\square$

## D. Equivalence of Policy Models and Induced Plackett-Luce (PL) Ranking Models

We consider two policy models: $\pi_\theta$ is a trainable policy model with parameters $\theta$, and $\pi^*$ is the optimal policy model. Recall that for a collection of trajectories $\mathcal{T} := \{\tau_1, \tau_2, \ldots, \tau_{|\mathcal{T}|}\}$ and query $\boldsymbol{x}$, the Plackett-Luce (PL) ranking model induced by a policy $\pi$, relative to a previous policy $\pi_{\text{old}}$, is defined as:

$$
P_\pi(\sigma | \mathcal{T}, \boldsymbol{x}) = \prod_{i=1}^{|\mathcal{T}|} \frac{\exp \left( \beta \log \frac{\pi(\tau_{\sigma(i)} | \boldsymbol{x})}{\pi_{\text{old}}(\tau_{\sigma(i)} | \boldsymbol{x})} \right)}{\sum_{j=i}^{|\mathcal{T}|} \exp \left( \beta \log \frac{\pi(\tau_{\sigma(j)} | \boldsymbol{x})}{\pi_{\text{old}}(\tau_{\sigma(j)} | \boldsymbol{x})} \right)}, \tag{20}
$$

where $\sigma$ is a permutation (or ranking) of the indices $\{1, 2, \ldots, |\mathcal{T}|\}$ and $\tau_{\sigma(i)}$ denotes the trajectory ranked at position $i$ in permutation $\sigma$. $\frac{\pi(\tau | \boldsymbol{x})}{\pi_{\text{old}}(\tau | \boldsymbol{x})}$ represents the relative preference of policy $\pi$ over the previous policy before updating. The following proposition supports the conclusion from Proposition 4.1.

**Proposition D.1** (Reward-shift invariance and policy-PL equivalence). *Fix $\beta > 0$ and suppose $\pi_{old}(\tau \,|\, \boldsymbol{x}) > 0$ on the support for every $\boldsymbol{x}$. Let $\pi^*$ denote the optimal policy of the KL-regularized RL problem in Eq. 5, with the closed-form structure in Eq. 6, and let $P_\pi(\sigma \,|\, \mathcal{T}, \boldsymbol{x})$ be the induced PL ranking model defined in Section D. Then:*

1. *(Reward-shift invariance) If $r'(\boldsymbol{x}, \tau) = r(\boldsymbol{x}, \tau) + \zeta(\boldsymbol{x})$ for some function $\zeta(\cdot)$ on the query $\boldsymbol{x}$, then the corresponding optimal policy is unchanged and so is its induced PL model; i.e., $\pi_{r'}^*(\tau \,|\, \boldsymbol{x}) = \pi_r^*(\tau \,|\, \boldsymbol{x})$ and $P_{\pi_{r'}^*}(\sigma \,|\, \mathcal{T}, \boldsymbol{x}) = P_{\pi_r^*}(\sigma \,|\, \mathcal{T}, \boldsymbol{x})$ for all $\sigma, \mathcal{T}, \boldsymbol{x}$.*

2. *(Policy–PL equivalence) For any learnable policy $\pi_\theta$,*

$$
P_{\pi_\theta}(\sigma \,|\, \mathcal{T}, \boldsymbol{x}) = P_{\pi^*}(\sigma \,|\, \mathcal{T}, \boldsymbol{x}) \text{ for all } \sigma, \mathcal{T}, \boldsymbol{x} \quad \Longleftrightarrow \quad \pi_\theta(\tau \,|\, \boldsymbol{x}) = \pi^*(\tau \,|\, \boldsymbol{x}) \text{ for all } \tau, \boldsymbol{x}.
$$

*Proof.* **Reward-shift invariance.** By Eq. 6, following Rafailov et al. (2023), for each fixed $\boldsymbol{x}$,

$$
\pi_r^*(\tau \,|\, \boldsymbol{x}) = \frac{\pi_{\text{old}}(\tau \,|\, \boldsymbol{x}) \exp\left(\frac{r(\boldsymbol{x}, \tau)}{\beta}\right)}{\sum_{\tau'} \pi_{\text{old}}(\tau' \,|\, \boldsymbol{x}) \exp\left(\frac{r(\boldsymbol{x}, \tau')}{\beta}\right)}.
$$

If $r'(\boldsymbol{x}, \tau) = r(\boldsymbol{x}, \tau) + \zeta(\boldsymbol{x})$, then

$$
\pi_{r'}^*(\tau \,|\, \boldsymbol{x}) = \frac{\pi_{\text{old}}(\tau \,|\, \boldsymbol{x}) \exp\left(\frac{r(\boldsymbol{x}, \tau) + \zeta(\boldsymbol{x})}{\beta}\right)}{\sum_{\tau'} \pi_{\text{old}}(\tau' \,|\, \boldsymbol{x}) \exp\left(\frac{r(\boldsymbol{x}, \tau') + \zeta(\boldsymbol{x})}{\beta}\right)}.
$$

The factor $\exp(\zeta(\boldsymbol{x})/\beta)$ cancels between numerator and denominator, hence $\pi_{r'}^*(\tau \mid \boldsymbol{x}) = \pi_r^*(\tau \mid \boldsymbol{x})$ for all $\tau, \boldsymbol{x}$. Since $P_\pi(\sigma \mid \mathcal{T}, \boldsymbol{x})$ depends on $\pi$ only through the ratios $\pi(\tau \mid \boldsymbol{x})/\pi_{\text{old}}(\tau \mid \boldsymbol{x})$, equality of the policies immediately implies $P_{\pi_{r'}^*}(\sigma \mid \mathcal{T}, \boldsymbol{x}) = P_{\pi_r^*}(\sigma \mid \mathcal{T}, \boldsymbol{x})$ for all $\sigma, \mathcal{T}, \boldsymbol{x}$.

**Policy and PL ranking equivalence.** The reverse direction ($\Leftarrow$) is straightforward: if $\pi_\theta = \pi^*$, then all ratios $\pi_\theta(\tau \mid \boldsymbol{x})/\pi_{\text{old}}(\tau \mid \boldsymbol{x})$ equal the corresponding ratios for $\pi^*$, so $P_{\pi_\theta} = P_{\pi^*}$ by definition.

For the forward ($\Rightarrow$) direction, suppose $P_{\pi_\theta}(\sigma \mid \mathcal{T}, \boldsymbol{x}) = P_{\pi^*}(\sigma \mid \mathcal{T}, \boldsymbol{x})$ for all $\sigma, \mathcal{T}, \boldsymbol{x}$. Fix any $\boldsymbol{x}$ and any two trajectories $\tau, \tau'$, and consider $\mathcal{T} = \{\tau, \tau'\}$. For this two-item set, the PL definition gives

$$\frac{P_\pi((\tau, \tau') \mid \{\tau, \tau'\}, \boldsymbol{x})}{P_\pi((\tau', \tau) \mid \{\tau, \tau'\}, \boldsymbol{x})} = \frac{\exp\left(\beta \log \frac{\pi(\tau|\boldsymbol{x})}{\pi_{\text{old}}(\tau|\boldsymbol{x})}\right)}{\exp\left(\beta \log \frac{\pi(\tau'|\boldsymbol{x})}{\pi_{\text{old}}(\tau'|\boldsymbol{x})}\right)} = \left(\frac{\pi(\tau \mid \boldsymbol{x})/\pi_{\text{old}}(\tau \mid \boldsymbol{x})}{\pi(\tau' \mid \boldsymbol{x})/\pi_{\text{old}}(\tau' \mid \boldsymbol{x})}\right)^\beta.$$

Applying this identity to $\pi = \pi_\theta$ and $\pi = \pi^*$ and using the supposed equality of PL models yields

$$\left(\frac{\pi_\theta(\tau \mid \boldsymbol{x})/\pi_{\text{old}}(\tau \mid \boldsymbol{x})}{\pi_\theta(\tau' \mid \boldsymbol{x})/\pi_{\text{old}}(\tau' \mid \boldsymbol{x})}\right)^\beta = \left(\frac{\pi^*(\tau \mid \boldsymbol{x})/\pi_{\text{old}}(\tau \mid \boldsymbol{x})}{\pi^*(\tau' \mid \boldsymbol{x})/\pi_{\text{old}}(\tau' \mid \boldsymbol{x})}\right)^\beta.$$

Since $\beta > 0$ and all terms are positive (by $\pi_{\text{old}} > 0$ on the support), taking the $1/\beta$ power gives

$$\frac{\pi_\theta(\tau \mid \boldsymbol{x})/\pi_{\text{old}}(\tau \mid \boldsymbol{x})}{\pi_\theta(\tau' \mid \boldsymbol{x})/\pi_{\text{old}}(\tau' \mid \boldsymbol{x})} = \frac{\pi^*(\tau \mid \boldsymbol{x})/\pi_{\text{old}}(\tau \mid \boldsymbol{x})}{\pi^*(\tau' \mid \boldsymbol{x})/\pi_{\text{old}}(\tau' \mid \boldsymbol{x})} \quad \Longleftrightarrow \quad \frac{\pi_\theta(\tau \mid \boldsymbol{x})}{\pi^*(\tau \mid \boldsymbol{x})} = \frac{\pi_\theta(\tau' \mid \boldsymbol{x})}{\pi^*(\tau' \mid \boldsymbol{x})}.$$

Thus, for each fixed $\boldsymbol{x}$, the ratio $\pi_\theta(\tau \mid \boldsymbol{x})/\pi^*(\tau \mid \boldsymbol{x})$ is constant in $\tau$. Summing over $\tau$ and using that both policies normalize to 1 implies this constant must be 1, hence $\pi_\theta(\tau \mid \boldsymbol{x}) = \pi^*(\tau \mid \boldsymbol{x})$ for all $\tau, \boldsymbol{x}$, which completes the proof.

$\square$

Proposition D.1 also establishes that our learning objective is invariant to any baseline reward adjustment, which is constant across all possible trajectories $\tau$ for a given query $\boldsymbol{x}$. This property is crucial as it ensures that our method learns the true *relative preferences* among demonstration sequences, which is the core of the selection task.

**Intuition.** The invariance property stems directly from the exponential form of the optimal policy solution in Eq. 6. Given $\pi^*(\tau|\boldsymbol{x}) \propto \pi_{\text{old}}(\tau|\boldsymbol{x}) \exp\left(r(\boldsymbol{x}, \tau)/\beta\right)$, if we use an equivalent reward $r'(\boldsymbol{x}, \tau) = r(\boldsymbol{x}, \tau) + \zeta(\boldsymbol{x})$, the new un-normalized policy becomes:

$$\pi'_{\text{un-normalized}}(\tau|\boldsymbol{x}) \propto \pi_{\text{old}}(\tau|\boldsymbol{x}) \exp\left(\frac{r(\boldsymbol{x}, \tau) + \zeta(\boldsymbol{x})}{\beta}\right) = \left(\pi_{\text{old}}(\tau|\boldsymbol{x}) \exp\left(\frac{r(\boldsymbol{x}, \tau)}{\beta}\right)\right) \cdot e^{\zeta(\boldsymbol{x})/\beta}$$

When this expression is normalized over all trajectories $\tau$ to compute the final policy, the term $e^{\zeta(\boldsymbol{x})/\beta}$ is a constant factor that appears in both the numerator and the denominator (the partition function) and thus cancels out. This leaves the final policy unchanged. Similar logic applies to the PL model, where the scores for all trajectories are scaled by the same factor, resulting in an identical probability distribution over rankings.

**Practical Implications.** This invariance is highly valuable in practice. It provides the flexibility to shape the reward function to incorporate additional context without distorting the underlying learning problem of ranking trajectories. For example, in a production environment, a practitioner could define a cost-aware reward $r'(\boldsymbol{x}, \tau) = r(\boldsymbol{x}, \tau) - \text{cost}(\boldsymbol{x})$, where $r(\boldsymbol{x}, \tau)$ is the performance reward from Eq. 4 and $\text{cost}(\boldsymbol{x})$ is a penalty based on query complexity (Chen et al., 2023). This result guarantees that adding this query-dependent cost term $\zeta(\boldsymbol{x}) = -\text{cost}(\boldsymbol{x})$ does not change the optimal policy's preference for one demonstration sequence over another *for that query*. Therefore, our approach of training $\pi_\theta$ to match $\pi^*$ remains a robust strategy focused on learning the optimal relative ordering of demonstration sequences.

