# OpenReview forum: "Auto-regressive In-context Demonstration Selection"
_ICML.cc/2026/Conference — ICML 2026 regular_

### Official Review · Reviewer_SWhi · 2026-02-19

**Soundness:** 3
**Presentation:** 2
**Significance:** 3
**Originality:** 2
**Overall Recommendation:** 5
**Confidence:** 4

**Summary:**

This paper addresses the problem of in-context demonstration selection for few-shot learning with LLMs. The key challenge is that demonstration selection is sensitive to both content (which examples to select) and ordering (the sequence of examples), and these two factors are interdependent.
The authors propose AUTOSELECT, an auto-regressive framework that frames demonstration selection as a sequential decision process. Unlike prior work that treats selection as independent retrieval, AUTOSELECT constructs demonstration sequences one step at a time, conditioned on the query and previous selections—similar to how LLMs generate text token by token. The method uses a ViT-based policy model operating on 2D matrix embeddings to preserve token structure, incorporates an adaptive stopping mechanism (EOS signal) to dynamically determine optimal sequence length, and optimizes the policy via a theoretically-grounded cross-entropy objective that minimizes an upper bound on the discrepancy to the optimal Plackett-Luce ranking.

**Compliance With Llm Reviewing Policy:**

Affirmed.

**Final Justification:**

The rebuttal has addressed my concern

**Key Questions For Authors:**

1. The experiments use pools of about 100 exemplars. How does the method scale to much larger pools (e.g., thousands of examples)？

2. The EOS signal is used to dynamically determine the length of the example sequence, but Figure 2 shows that the EOS probability distribution varies greatly across different tasks (e.g., the EOS probability is significantly lower for the Subj task). Does this mean that the model has difficulty determining when to stop on certain tasks?

3. The training phase requires generating K complete sequences for each query and calling LLM to evaluate the reward, which is costly for large-scale datasets or expensive APIs. The paper mentions "offline training, one-time investment" but does not report the specific training overhead.

**Strengths And Weaknesses:**

Strength:

+ The auto-regressive framing unifies content selection and ordering into a single tractable framework, rather than treating them as separate problems or ignoring ordering effects.
+ The adaptive stopping mechanism (EOS signal) allows dynamic sequence length determination, and the method is trained offline as a one-time investment, making inference efficient.

Weakness:

+ While inference is efficient, the training requires generating K rollouts per query and evaluating rewards via LLM queries, which may be expensive for large-scale deployment.
+ The reward is defined only at the sequence level (final accuracy), which may provide limited signal for credit assignment during early steps of the sequence.

---

> ### Author Rebuttal · Authors · 2026-03-31
>
> We sincerely thank the reviewer for the careful reading and constructive feedback. We will be glad to address any further questions or concerns.
>
>
> ---
> > Q1. How does the method scale to larger exemplar pools?
> ---
>
>
>
> - We agree that scaling to much larger pools is an important scenario. In the current implementation, each decision step scores all remaining candidates plus EOS, so the per-step selection cost grows roughly linearly with the current pool size.
> In this case, our Table 5 in the Appendix shows that AutoSelect scales **mildly** with larger exemplar pools: selection time increases from roughly 6-7 ms per step at pool size 100 to only 31-34 ms at pool size 1000, consistent with the $O(TN)$ computing cost. This overhead remains mild, so even with larger pools the method remains practically efficient within our few-shot setting.
>
>
>
>
> ---
> > Q2. Figure 2 shows that EOS probability varies a lot across tasks. Does this mean the model has difficulty deciding when to stop?
> ---
>
> - We would like to note that **Fig. 10 is essentially the evidence of our task-dependent adaptive stopping behavior that AutoSelect aims to achieve.**
>
> - Different tasks benefit from different context lengths, so a lower EOS probability on one task means that the learned policy tends to prefer slightly longer demonstration sequences there. **This is exactly the purpose of our EOS design.** The paper also shows that AutoSelect uses about three exemplars on average rather than always saturating the maximum length, which supports the view that **the policy is learning when additional context is useful, rather than blindly filling the budget.**
>
>     - This also ensures efficient usage of the LLM context window by achieving strong performance with a reasonable amount of exemplars.
>
>
>
>
>
> ---
> > Q3. Breakdown of the offline training?
> ---
>
>
> - Our Fig. 5 shows that AutoSelect achieves the strongest overall performance-efficiency trade-off among learning-based baselines: it consistently attains the highest normalized performance while requiring reasonable running costs. Thus, the one-time offline cost of AutoSelect is not only practically modest, but also more favorable overall than competing learning-based selectors when both quality and runtime are considered together.
>
>
> - We agree that the offline cost should be quantified explicitly. Under our standard AutoSelect setting with Qwen2.5-3B, each run performs 1200 rollouts and around 2.53 reward evaluations per rollout on average.
>
> - Across the datasets, 18.1\% of end-to-end runtime is spent in actual downstream LLM calls and the remaining 81.9\% is selector-side/model-side computation. Within the LLM-call budget, offline reward evaluation accounts for 51.9\%, validation checks during training for 43.3\%, and the final held-out test for only 4.8\%. We believe this one-time offline cost is worthwhile because it learns a reusable selector, with only modest test-time inference overhead (e.g., 0.282s/query for AutoSelect vs. 0.212s/query for BM25), including exemplar selection and consequent LLM inference.
>
>
>
> ---
> > Q4. Why is the reward only sequence-level? Weak credit-assignment signal?
> ---
>
> Our sequence-level reward design is intentional because it is exactly what allows the method to capture **compositional effects** of content and ordering without requiring expensive exemplar-level labels.
>
> If we instead optimize exemplar-level scores, we can lose the ability to directly train against the downstream effect of a whole ordered demonstration sequence. AutoSelect addresses this by sampling multiple ordered trajectories during training, optimizing the sequence-level objective induced by downstream task reward, and stabilizing learning through replay in an efficient way.
>
> The sequence-level signals are also common for existing learning-based baselines (e.g., Wu et al., 2024) in our targeted few-shot demonstration selection research direction.

---

> > ### Author Rebuttal · Reviewer_SWhi · 2026-04-03
> >
> > Thank you for your response，I will increase my score

---

> > > ### Author Response · Authors · 2026-04-03
> > >
> > > We sincerely thank the reviewer for the supportive follow-up and truly appreciate the score increase! We are very glad that our rebuttal has adequately addressed your concerns, and we have incorporated our discussion into the revised manuscript.

---

### Official Review · Reviewer_pYni · 2026-02-24

**Soundness:** 3
**Presentation:** 3
**Significance:** 2
**Originality:** 3
**Overall Recommendation:** 5
**Confidence:** 4

**Summary:**

This paper proposes AutoSelect, a framework that formulates few-shot in-context demonstration selection as an auto-regressive sequential decision process. At each step, a trainable policy model selects the next exemplar conditioned on the query and previously chosen demonstrations. The framework formulates a KL-regularized RL objective whose optimal policy induces a Plackett-Luce ranking over demonstration sequences. The authors prove that minimizing a policy-level Cross-Entropy loss bounds the worst-case discrepancy between the learned and optimal PL rankings. Empirically, AutoSelect is evaluated on nine datasets and achieves up to 11% improvement over the strongest baseline on challenging reasoning tasks.

**Compliance With Llm Reviewing Policy:**

Affirmed.

**Final Justification:**

The authors have helped me clear up my misunderstanding.

**Key Questions For Authors:**

1. Could the reward function be extended to handle ROUGE, BLEU, or generally other metrics?
2. Why was [1] excluded from the baseline comparisons, and how do you expect your method would perform against it given its relevance to your approach?
3. Can you provide a cost analysis comparing the LLM inference calls required for reward evaluation against retrieval-based alternatives, particularly as the number of in-context demonstrations scales beyond T=4?

**Limitations:**

Yes.

**Strengths And Weaknesses:**

## Strengths
- The auto-regressive formulation for demonstration selection is well-motivated and the proposed method is original, with clear theoretical grounding.
- Gains on hard tasks are substantial and universal across model sizes.
- The proposed method is compared against strong baselines.

## Weaknesses
- The authors cite [1] but does not include it as part of baseline.
- All dataset evaluated on are short-form generation tasks. Showing performance gain on "generative" tasks would substantially boost the work's significance.
- The method is evaluated primarily at T=4 with extensions to T=16 and T=24. This could be a major drawback as many-shot ICL are getting more and more popular. The authors admit this weakness.
- LLM inference calls for reward evaluation is still substantially more complex than retrieval-based methods. The authors also did not include a cost analysis for this part of the pipeline.

I am setting my score at 3 tentatively. If authors can address the first two weaknesses, I am happy to raise my score.

[1] Wang et al., "Demonstration Selection for In-Context Learning via Reinforcement Learning", ICML 2025

---

> ### Author Rebuttal · Authors · 2026-03-31
>
> We sincerely thank the reviewer for the careful reading and constructive feedback. We will be glad to address any further questions or concerns.
>
>
> ---
> > Q1. Could AutoSelect be extended to ROUGE, BLEU? Generative tasks?
> ---
>
> - Our AutoSelect is not tied to the accuracy metric. In Eq. (4), the reward is defined as $r(x,\tau)=L(y,\mathrm{LLM}(x;\tau))$, where $L(\cdot,\cdot)$ is simply the task metric. This formulation supports ROUGE, BLEU or other task-specific metrics.
>
>
> - Here, we verified this on two generative tasks using Qwen3-8B ($T=4$) on "xsum" [1] (an abstractive news summarization task evaluated by ROUGE-L) and "e2e\_nlg" [2] (a data-to-text generation task evaluated by BLEU).
>
>
> | Dataset (Metric) | AutoSelect | BM25 | Random |
> |---|---:|---:|---:|
> | xsum (ROUGE-L) | 0.2015 | 0.1886 | 0.1832 |
> | e2e\_nlg (BLEU) | 0.2614 | 0.2400 | 0.2361 |
>
>
> AutoSelect remains competitive and consistently improves over baselines even when the rewards are generation metrics.
>
> [1] Narayan, et al. "Don’t give me the details, just the summary! topic-aware convolutional neural networks for extreme summarization."
>
> [2] Novikova, et al. "The E2E dataset: New challenges for end-to-end generation."
>
>
>
> ---
> > Q2. Why was RDES (Wang et al.; 2025) excluded from the baseline comparisons?
> ---
>
> - To the best of our knowledge, there is no official code implementation published for RDES (Wang et al. (2025)). We thus did not include a direct empirical comparison in the first place.
>
> - To address the reviewer's question, we implement RDES ourselves, and adapt it to our settings for fair comparisons. We run RDES for three seeds and report mean performance on Qwen2.5-3B, where AutoSelect keeps achieving better performance.
>
>
> | Method \ Task | Winowhy | Epistemic\_reasoning | Aqua | Trec |
> | :--- | :---: | :---: | :---: | :---: |
> | RDES            | 0.577 | 0.525 | 0.375 | 0.382 |
> | CEIL            | 0.591 | 0.546 | 0.344 | 0.375 |
> | AutoSelect      | 0.657 | 0.601 | 0.395 | 0.393 |
>
>
> ---
> > Q3. How should readers interpret this with respect to many-shot ICL?
> ---
>
>
> - We would like to mention that our main focus is on the few-shot regime (Zhang et al., 2022; Wu et al., 2024; Purohit et al., 2025), not extreme many-shot ICL. In this case, the goal of few-shot selection is not to merely maximize accuracy, but to achieve the **highest possible accuracy within a demonstration budget**, which is the core *accuracy-cost trade-off at inference time*.
>
>     - As noted in previous works (Purohit et al., 2025), the few-shot exemplar setting is often treated as the  **efficiency sweet spot** for moderate-sized LLMs, balancing strong performance gains with low computational overhead in terms of the context window of LLM inference.
>
>
> - In particular, for Transformer-based LLMs with large numbers of parameters, the computational complexity of the attention mechanism **scales quadratically** ($O(n^2)$) with the input sequence length $n$.
>
> - *This means that for a large-scale application with a high volume of inference queries, the LLM inference cost of many-shot prompts can become overly expensive.* Our work, and others in this few-shot research direction, operate on the premise that finding a small sequence of exemplars is more practical and cost-effective than defaulting to a "many-shot" prompt for every task. The quality, not just the quantity, of these exemplars is paramount, as performance is highly sensitive to the specific demonstrations provided.
>
>
> - On the other hand, the performance on **considerably longer** sequences (e.g., $T \gg 16$) pertains to long-context RAG, which corresponds to a different research direction and is a **significantly different problem setting** from our work's focus on few-shot ICL demonstration selection.
>     - *We also have Appendix B.3 to discuss extending our auto-regressive formulation to long-context RAG (e.g., with adaptations like sparse attention mechanisms).* We identify this as a valuable direction for future research and **outline potential technical pathways** for this extension.
>
>
> ---
> > Q4. Can you provide a cost analysis versus retrieval-based methods?
> ---
>
>
> Thank you for the suggestion. We add an inference time comparison against retrieval-based BM25 under Qwen2.5-3B. BM25 requires $0.212 / 0.401 / 0.627 / 0.935$ seconds per query at $T=4/10/16/24$, including exemplar selection and consequent LLM inference. AutoSelect requires $0.282 / 0.546 / 0.841 / 1.162$ seconds per query at the same shot counts. Thus, the overhead is moderate, while both methods exhibit the same monotonic increase with context length. Meanwhile, at deployment, AutoSelect does not perform reward evaluation, and inference consists only of selector-based exemplar choice plus the same downstream LLM prediction step used by retrieval baselines. We have included this cost analysis, and clarified the distinction between training-time reward-evaluation overhead and deployment-time inference cost in the revision.

---

> > ### Author Rebuttal · Reviewer_pYni · 2026-03-31
> >
> > I have raised my score.

---

> > > ### Author Response · Authors · 2026-03-31
> > >
> > > Thank you very much for your thoughtful review and for taking the time to revisit our rebuttal! We sincerely appreciate your careful consideration of our responses, and we are grateful that our clarifications were able to address your concerns.

---

### Official Review · Reviewer_45Mq · 2026-03-11

**Soundness:** 2
**Presentation:** 3
**Significance:** 3
**Originality:** 2
**Overall Recommendation:** 3
**Confidence:** 4

**Summary:**

This paper studies demonstration selection and ordering in few-shot in-context learning, and proposes AUTOSELECT, which formulates the problem as an auto-regressive sequential decision process. The topic is relevant, and the overall framework is reasonably well organized. The paper also includes additional experiments in the appendix beyond the main results.

**Compliance With Llm Reviewing Policy:**

Affirmed.

**Final Justification:**

Thank you again for the thoughtful and timely follow-up. While I still feel that some of my original concerns have not been fully resolved, I do appreciate the authors' effort in responding carefully and in trying to strengthen the paper during the rebuttal period.

In particular, I appreciate the authors' timely engagement, the additional clarifications, and the overall effort they have put into this work. Although I still have some reservations, I do think the discussion has been helpful and has improved my understanding of the paper and its intended scope.

Taking these points into account, I increase my score by one point.

I hope the authors will consider incorporating the main clarifications from the rebuttal into a future revision of the paper, especially those related to the scope of the work and the practical settings.

**Key Questions For Authors:**

1. Would the main conclusions still hold if the primary experiments were conducted on stronger backbones rather than mainly on Qwen2.5-3B?

2. Can the authors provide a clearer quantitative breakdown of total training cost, including rollouts, reward evaluations, and LLM calls?

3. Do the authors have stronger evidence that the method scales beyond the current few-shot regime toward many-shot in-context learning?

**Limitations:**

No. I do not think the paper discusses its limitations sufficiently clearly. In particular, the authors should be more explicit that the current evidence is still mainly limited to few-shot settings, that the main results rely on a relatively weak backbone, and that the cost-benefit trade-off is not yet fully established.

**Strengths And Weaknesses:**

## Strengths

1. **The problem is meaningful.**
   The paper does not treat demonstration selection as a static ranking problem, but tries to jointly model query dependence, ordering effects, and compositional interactions. This is a worthwhile direction.

2. **The experimental scope is fairly broad.**
   In addition to the main table, the paper includes extra comparisons and supplementary analyses in the appendix, which makes the empirical section more complete than many papers in this area.

## Weaknesses

1. **The main results still rely primarily on a relatively weak 3B backbone.**
   The main experiments are still centered on Qwen2.5-3B. Although the appendix adds stronger backbones, the core conclusions are not primarily established on them.

2. **The cost analysis is still not sufficiently convincing.**
   The paper repeatedly describes training as a one-time offline cost, but does not clearly quantify the total training cost in terms of rollouts, reward evaluations, or LLM calls. This makes it hard to judge the real practical trade-off.

3. **The scalability to many-shot in-context learning is still not properly validated.**
   Even with the appendix experiments, the paper remains mainly focused on few-shot settings. I do not think the current evidence is enough to support stronger claims beyond that scope.

4. **The gains are modest on several standard tasks.**
   While the method performs better on some harder reasoning tasks, the improvements on several classification benchmarks remain small. This makes the extra complexity harder to justify in practice.

5. **Some of the efficiency comparisons are not fully fair.**
   For example, comparing 4-shot AUTOSELECT with 16-shot random changes both the method and the context length at the same time, so it does not cleanly isolate the efficiency of the proposed method itself. A fairer comparison would fix the number of shots across methods or compare methods at matched performance levels.

---

> ### Author Rebuttal · Authors · 2026-03-31
>
> We sincerely thank the reviewer for the careful reading and constructive feedback. We will be glad to address any further questions or concerns.
>
>
> ---
> > Q1. Other backbones beyond Qwen2.5-3B?
> ---
>
> - **We validate our method's generalizability in the Appendix B.1**, showing AutoSelect's consistently strong performance **across various LLM families (e.g., GPT, Qwen and Llama) and scales (e.g., 8B and 14B models)**.
>
>     - For Table 1, we apply the Qwen2.5-3B model as a controlled base to ensure a **fair and direct comparison** of the *selection methods themselves*, across multiple runs.
>
> - To further address reviewer's concern, **we also add experiments on Qwen3-8B model**, where AutoSelect can still offer strong performance:
>
>
> | Method \ Task | Winowhy | Epistemic\_reasoning | Aqua | Trec |
> | :--- | :---: | :---: | :---: | :---: |
> | Random                    | 0.617 | 0.793 | 0.424 | 0.364 |
> | Retrieval-based Top-k     | 0.661 | 0.827 | 0.451 | 0.412 |
> | AutoSelect                | 0.703 | 0.897 | 0.475 | 0.483 |
>
>
>
> ---
> > Q2. Quantitative breakdown of rollouts, reward evaluations, and LLM calls?
> ---
>
>
> - Our Fig. 5 shows that AutoSelect achieves the strongest overall performance-efficiency trade-off among learning-based baselines: it consistently attains the highest normalized performance while requiring reasonable running costs. Thus, the one-time offline cost of AutoSelect is not only practically modest, but also more favorable overall than competing learning-based selectors, when both quality and runtime are considered together.
>
>
> - Under our standard AutoSelect setting with Qwen2.5-3B, each run performs 1200 rollouts and around 2.53 reward evaluations per rollout on average.
>
>
> - Across the datasets, 18.1\% of end-to-end runtime is spent in actual downstream LLM calls and the remaining 81.9\% is selector-side/model-side computation. Within the LLM-call budget, offline reward evaluation accounts for 51.9\%, validation checks during training for 43.3\%, and the final held-out test for only 4.8\%. We believe this one-time offline cost is worthwhile because it learns a reusable selector, with only modest test-time inference overhead (e.g., 0.282s/query for AutoSelect vs. 0.212s/query for BM25), including exemplar selection and consequent LLM inference.
>
>
> ---
> > Q3. Many-shot ICL?
> ---
>
> - We thank the reviewer for this important question. Our work is aimed at **few-shot demonstration selection** (Zhang et al., 2022; Wu et al., 2024; Purohit et al., 2025), where the goal is to maximize accuracy under a limited demonstration budget. *The few-shot exemplar setting is often treated as the  **efficiency sweet spot** for moderate-sized LLMs (Purohit et al., 2025).*
>
> - In this regime, selecting a small, high-quality exemplar sequence is often more useful than simply increasing the number of demonstrations, since LLM inference cost grows rapidly with prompt length. *Accordingly, we do not position AutoSelect for extreme many-shot settings, which are closer to long-context RAG and constitute a significantly different problem.*
>
> - Due to page limit, we gently refer the reviewer to a more comprehensive discussion of the many-shot settings in our response to Q3 of reviewer "pYni".
>
>
>
> ---
> > Q4. The gains are modest on some tasks?
> ---
>
>
> - We agree that gains on some datasets can be modest; **their main role is to show that AutoSelect remains stable and competitive across a broad range of tasks, including less difficult ones.**
>
>     - *The main evidence for AutoSelect instead comes from harder **reasoning-intensive tasks**, where order-aware and query-aware composition is more important.*
>
> - Due to page limit, we gently refer the reviewer to our detailed response to Q1 of reviewer "iK2R", **including additional results on two new generative tasks (XSum and E2E), showing that AutoSelect keeps delivering strong performance.**
>
>
> ---
> > Q5. Comparing 4-shot AutoSelect with 16-shot random baseline in Table 11?
> ---
>
> - We would like to clarify that *the comparison in Table 11 is intentionally asymmetric*:
>
>     - Its goal is not to isolate a single variable in a controlled ablation, but to show a practical outcome that *a learned 4-shot sequence selected by AutoSelect can simultaneously **achieve better performance** and **lower inference cost** than a much longer 16-shot sequence selected by Random.*
>
> - Thus, **Table 11 should be interpreted as an application-oriented efficiency comparison**, illustrating that better exemplar selection can reduce prompt length (i.e., LLM inference cost) while improving accuracy.
>
> - Our primary empirical evidence comes from matched-shot comparisons in the main paper and appendix, where AutoSelect and baselines are evaluated under the same candidate pool, supervision protocol, and trajectory-length setting (Table 1; Appendix B.1--B.2).
>
>     - **Table 1 also already includes the direct 4-shot comparison between AutoSelect and baselines including Random.**

---

> > ### Author Rebuttal · Reviewer_45Mq · 2026-04-04
> >
> > Thank you for the detailed rebuttal and for adding new results. I appreciate the effort the authors put into addressing my questions. That said, after reading the response, I still think several of my main concerns remain only partially addressed.
> >
> > First, the additional Qwen3-8B results are useful, but I am still not fully convinced that the main conclusions would remain equally strong if the core evidence of the paper were based on stronger backbones rather than mainly on Qwen2.5-3B. Part of my concern here is also that Qwen2.5-3B now feels somewhat dated for a 2026 evaluation standard, since it is a relatively old and small backbone. The rebuttal gives some support for generality, but not enough to fully remove the concern that the main empirical case is still anchored on an older and relatively weak model.
> >
> > Second, I still find the cost analysis not sufficiently clear. My original concern was about a clearer quantitative accounting of the overall training cost, especially the total number of LLM calls and the corresponding training cost. The rebuttal provides some percentages and average statistics, but it is still difficult for me to judge the absolute end-to-end training cost and, therefore, the practical cost-effectiveness trade-off.
> >
> > Third, I do not think my concern about many-shot scalability was directly answered. My question was whether the current evidence is sufficient to support claims beyond few-shot settings. The rebuttal instead argues that extreme many-shot settings are closer to long-context RAG. I do not find this fully convincing. Many-shot ICL and RAG are still different settings: the former is still based on demonstrations, while the latter relies on retrieved external evidence. So to me, this part of the response shifts the discussion to a related but different problem, rather than directly addressing the original concern.
> >
> > I also appreciate the clarification that some datasets are mainly included to show stability. At the same time, this seems to reinforce the point that the added complexity may be easier to justify on more reasoning-intensive tasks than on standard tasks where the gains are relatively small. I think this limitation should be stated more explicitly.
> >
> > For the comparison between 4-shot AutoSelect and 16-shot random, I understand the authors' explanation that this is meant as an application-oriented efficiency comparison rather than a controlled ablation. That is a fair clarification. Still, I think the framing should be careful, since both the method and the context length are changed at the same time, so this comparison alone cannot establish a fully fair method-level efficiency advantage.
> >
> > Overall, I appreciate the authors' clarifications and additional results. However, I am still not fully convinced that the rebuttal resolves the concerns about backbone strength, cost transparency, and support for claims beyond few-shot settings. I will take the rebuttal and the other reviewers' comments into account before deciding whether I should revise my score.

---

> > > ### Author Response · Authors · 2026-04-05
> > >
> > > We thank the reviewer for the follow-up comments. We are glad that most of your concerns have been addressed. For your remaining questions, we are happy to provide additional detailed responses below.
> > >
> > > ---
> > > > **On many-shot scope:**
> > > ---
> > >
> > >   - We would like to kindly clarify a potential misunderstanding here: throughout our paper, we do **not** claim validation beyond the fixed-budget **few-shot ICL** regime. Our target setting is precisely the one where only **a small number of exemplars** can be used, so the key challenge is **which** demonstrations to include and **how** to order them under a tight context budget.
> > >
> > >   - By contrast, the many-shot regime (e.g., [1]) is materially different in both scale and computational cost: [1] uses **Gemini 1.5 Pro with 1M-token context window**, evaluates **hundreds to thousands of shots**, and reports that best performance is often reached only at very large token budgets. This is therefore not a small extension of our setting, but a substantially different operating point with far larger context capacity and test-time budget. Our point is simply that **broad many-shot validity is not the claim of this paper**.
> > >
> > >   - We have revised the paper to state this scope explicitly, and presented broader many-shot evaluation as valuable future work in Appendix B.3. This clarification is also consistent with other reviewers' reactions to closely related concerns: `Reviewer pYni` raised a related many-shot question and later agreed with our response while also raising the score, and `Reviewer iK2R` likewise agreed that AutoSelect is appropriate for the targeted few-shot regime.
> > >
> > >   [1] Agarwal et al., *Many-Shot In-Context Learning*, NeurIPS 2024
> > >
> > >
> > >
> > >
> > > ---
> > > > **On the choice of backbones:**
> > > ---
> > >
> > >   - We agree that if the paper only reported Qwen2.5-3B, the evaluation would indeed be insufficient to generalize the advantage of AutoSelect. **That is exactly why we do not rely on Qwen2.5-3B alone**: Appendix B.1 includes broader **cross-backbone evidence with different LLM families and sizes**, and we further added new results on stronger Qwen3-8B in the rebuttal so that the empirical case is not anchored to Qwen2.5-3B.
> > >
> > >   - At the same time, using Qwen2.5-3B in the main table serves a specific methodological purpose. It is a capable and stable task-solving model that makes exhaustive **multi-dataset, multi-seed, controlled** comparisons feasible under a fixed budget, allowing Table 1 to isolate the quality of the **selector** rather than the absolute strength of a particular downstream LLM. In other words, the main table is designed for **controlled selector evaluation**.
> > >
> > >   - Accordingly, our claim is that AutoSelect's advantage is **not limited to a specific model architecture or capability level**, and the added cross-model results are included specifically to support that claim.
> > >
> > >
> > >
> > >
> > > ---
> > > > **On training cost transparency:**
> > > ---
> > >
> > >   - For one AutoSelect run, the full pipeline involves an average of 40k total task-LLM calls across training, evaluation, and testing, with 2.64 GPU hours on GH200.
> > >   For a direct comparison, our closest learning-based baseline in GPU hours, EASE, uses 50k LLM calls with 2.79 GPU hours. We also report deployment overhead at test time as 0.282 sec/query for AutoSelect, including exemplar selection and consequent LLM inference, versus 0.212 sec/query for the efficient retrieval baseline BM25. This makes the one-time offline cost and its trade-off against downstream inference savings directly measurable, rather than only relatively described.
> > >
> > > - `Reviewer pYni` and `Reviewer SWhi` also agreed with our response to these cost-related concerns after our quantitative clarifications.
> > >
> > >
> > >
> > >
> > > ---
> > > > **On gains on standard tasks and the 4-shot vs 16-shot comparison:**
> > > ---
> > >
> > >   We agree that the practical value of AutoSelect is most evident on reasoning-intensive tasks, where ordering-aware and query-aware exemplar composition has the largest effect, and we have updated the paper to state this more concretely in Subsec. 5.1.
> > >
> > >   - This interpretation is also consistent with `Reviewer iK2R`'s previous question about less difficult datasets. *The reviewer later agreed with our response after our clarification that the harder reasoning tasks provide the main evidence, while the broader suite serves as breadth and stability checks.*
> > >
> > >   - Consistent with this positioning, we have also revised the discussion of Table 5 to frame the 4-shot vs 16-shot result as an **application-oriented budget comparison**. Our main method-level evidence remains the matched-shot comparisons in the main paper and appendix.
> > >
> > >
> > >
> > > ---
> > > Thank you again for your thoughtful feedback and dedication to the review process.
> > > **We hope our response has helped clarify the scope of the paper, and we would be grateful if the reviewer could take both the rebuttal and the other reviewers' comments into account when reconsidering our submission.**

---

### Official Review · Reviewer_iK2R · 2026-03-13

**Soundness:** 3
**Presentation:** 3
**Significance:** 2
**Originality:** 4
**Overall Recommendation:** 4
**Confidence:** 4

**Summary:**

The paper introduces AutoSelect, an auto-regressive framework for selecting and ordering demonstrations for few-shot in-context learning. Given an input example and a pool of candidate demonstrations, the method first embeds both into a representation designed to preserve token-level structure.AutoSelect then uses a trainable policy to build a demonstration sequence, one example at a time. Each choice is conditioned on the input and on the demonstrations already selected. At each step, the policy can either select another demonstration or stop early, allowing it to adapt the number of demonstrations to the input. After generating a set of candidate sequences, each sequence is evaluated using a fixed downstream language model to obtain a sequence-level reward based on downstream task performance. The policy is trained using these rewards with a cross-entropy style objective defined over the sampled sequences. Across tasks, the results show that AutoSelect improves average performance relative to the baselines it evaluates, indicating that explicitly modeling demonstration order and interactions can help. The paper also provides a theoretical justification for the learning objective. It frames the goal as matching an optimal Plackett-Luce ranking distribution over demonstration-sequence orderings, and shows that minimizing the proposed cross-entropy objective minimizes an upper bound on the discrepancy from the optimal ranking.

**Compliance With Llm Reviewing Policy:**

Affirmed.

**Final Justification:**

The paper is interesting and proposes an interesting approach to ordering training samples for in-context learning. While there are some clear limitations, given that the paper focuses only on smaller label budgets, I still found its approach reasonable.

The rebuttal has reinforced my prior assessment. The authors have agreed to explicitly include the limitations in the paper.

**Key Questions For Authors:**

Please answer the following questions:

**Q1. Many-shot learning**
How does AutoSelect perform in a many-shot setting when the number of examples is greater than, say, 128 samples? Since the training itself is combinatorial in choosing trajectories, would it make sense to use AutoSelect in this setting?

**Q2. Quantifying the rollouts/trajectories.**
Would it be possible to quantify the number of rollouts you require to achieve the downstream performance that you’ve reported in Table 1?

**Q3. Oracle-based methods.**
The authors note that oracle-based methods are “significantly more costly than other baselines and AutoSelect”. Could you clarify how much more expensive the greedy-oracle is than AutoSelect? The paper does not provide a complete picture.

**Limitations:**

yes

**Strengths And Weaknesses:**

### Strengths

The paper is well written, and Figure 2 does a great job of explaining the overall training pipeline.
The AUTOSELECT framework is novel. It treats the ordering of demonstrations as a sequential decision process, which allows the authors to draw on ideas from reinforcement learning to learn a policy that orders examples for few-shot in-context learning.
The paper shows that AUTOSELECT improves downstream few-shot in-context learning performance, on average, compared to the evaluated baselines. In addition, the cross-task transfer experiment is interesting: the authors evaluate whether a policy trained on one task can be applied to new tasks, and they often report meaningful gains over random selection.

### Weaknesses

**Downstream datasets.**
The main limitation of this work is that the evaluation datasets are largely saturated; newer language models often show high performance on datasets such as AGNews, Amazon, and SST-2. While the paper shows improvements on these datasets, this significantly reduces the impact of the work. Furthermore, it is unclear why the paper evaluates on only four datasets from BIG-bench.

**Missing baseline.** The paper is missing a baseline where the downstream model is fine-tuned on the few-shot samples. At the moment, the task is somewhat artificial, since the amount of compute spent on learning the policy could potentially be used to train the downstream model on few-shot examples. Furthermore, prior work has found that fine-tuning is more beneficial than in-context learning [a]. It would strengthen the paper if it included a comparison of training FLOPs and downstream performance.

**Many-shot setting.**
Recent work has shown that in-context learning performance increases with more samples [b, c]. However, in the current setup, the number of few-shot samples is rather limited; the paper does not consider a setting with many in-context samples. Furthermore, this would require rolling out an enormous number of trajectories, which raises questions about the practical utility of the AutoSelect framework.

**Relevance of few-shot in-context learning.**
This concerns the paper’s potential impact. It is unclear whether few-shot in-context learning is particularly relevant, since post-trained language models can perform many tasks out of the box and achieve strong performance. It would be interesting to know in which settings this setup and pipeline are useful. Additionally, I am curious whether post-trained language models also benefit from this pipeline.


**References.**

[a] Few-Shot Parameter-Efficient Fine-Tuning is Better and Cheaper than In-Context Learning. NeurIPS 22.

[b] Many-shot in-context learning, NeurIPS 2024.

[c] In-context learning learns label relationships but is not conventional learning, ICLR 24.

---

> ### Author Rebuttal · Authors · 2026-03-31
>
> We sincerely thank the reviewer for the careful reading and constructive feedback. We will be glad to address any further questions or concerns.
>
> ---
> > Q1. Saturated datasets? Why four tasks of Big-bench?
> ---
>
> - We agree that gains on some datasets can be modest; **their main role is to show that AutoSelect remains stable and competitive across a broad range of tasks, including less difficult ones (e.g., AGNews, Amazon)**.
>
>     - *The main advantage for AutoSelect instead comes from harder reasoning-intensive settings, where order-aware and query-aware demonstrations are more important.*
>
>
> - **This is also why we include four relatively difficult BIG-bench reasoning tasks.** The larger gains appear on harder reasoning tasks, while the more saturated datasets mainly serve as breadth and stability checks; the harder reasoning tasks provide strong evidence for our method, and the broader suite shows that it is not brittle outside reasoning tasks.
>
>
> - In addition, **we add experiments on two generative tasks**, using Qwen3-8B ($T=4$) on "xsum" (a summarization task evaluated by ROUGE-L) and "e2e\_nlg" (a data-to-text generation task evaluated by BLEU), where AutoSelect remains strong.
>
> | Dataset (Metric) | AutoSelect | BM25 | Random |
> |---|---:|---:|---:|
> | xsum (ROUGE-L) | 0.2015 | 0.1886 | 0.1832 |
> | e2e\_nlg (BLEU) | 0.2614 | 0.2400 | 0.2361 |
>
>
>
>
> ---
> > Q2. Comparing with the fine-tuning baseline?
> ---
>
>
> - We would like to gently mention that the cited paper (T-Few) studies a distinct setting from ours. AutoSelect is designed for the fixed-downstream-model few-shot ICL regime (Zhang et al., 2022; Wu et al., 2024; Purohit et al., 2025), where the goal of this line of research is to improve **inference-time prompt construction**, by modeling query-aware ordering and compositional interactions among demonstrations, rather than updating the task-solving model itself. *This setting is well motivated when the downstream model is meant to remain fixed, such as in API-based scenarios.*
>
>
> - We also compare with T-Few by adapting it to Qwen models. On the Qwen2.5-3B, AutoSelect obtains 0.657 versus 0.644 for T-Few on "Winowhy", and also obtains 0.395 vs. 0.385 on AQuA. AutoSelect also outperforms T-Few on the new summarization task "e2e\_nlg" (0.2614 vs. 0.2402) with the Qwen3-8B. For model-training FLOPs averaged across these datasets, AutoSelect uses 9.20e14 FLOPs, compared with 2.62e15 for T-Few.
>
>
>
>
>
>
> ---
> > Q3. Can AutoSelect extend to many-shot settings?
> ---
>
>
> - We thank the reviewer for this important question. Our work aims at **few-shot demonstration selection** (Zhang et al., 2022; Wu et al., 2024; Purohit et al., 2025), where the goal is to maximize accuracy under a limited demonstration budget, since the few-shot exemplar setting is often treated as the  **efficiency sweet spot** for moderate-sized LLMs (Purohit et al., 2025).
>
> - In this regime, selecting a small, high-quality exemplar sequence is often more useful than simply increasing the number of demonstrations, since LLM inference cost grows rapidly with prompt length. Accordingly, we do not position AutoSelect for extreme many-shot settings, which are closer to long-context prompting / RAG and constitute a significantly different problem setting.
>
> - Due to page limit, we gently refer the reviewer to a more comprehensive discussion of the many-shot setting in our response to Q3 of reviewer "pYni".
>
>
>
> ---
> > Q4. Quantify the number of rollouts to achieve the reported performance?
> ---
>
> Under our experiment setup (Appendix A.3), our training uses 400 episodes with $K=3$ rollouts per episode, so each run uses 1200 rollouts.
> Meanwhile, our replay buffer enables the effective reuse of rollouts, which make them a mild offline cost, and the learned policy is efficient at deployment.
>
>
>
> ---
> > Q5. How expensive is greedy-oracle?
> ---
>
>
> AutoSelect one-time training (as in Appendix A.3) yields: 400 episodes $\times$ 3 rollouts $\times$ 5 aggregated samples $\approx$ 6,000 calls.
> In contrast, the deployment cost of Greedy-oracle for our test set ($T=4$) is: **400 test samples $\times$ (100+99+98+97) calls $\approx$ 157,600 calls.**
> This concrete trade-off justifies our claim of a "modest and one-time offline investment".
>
>
> ---
> > Q6. Can post-trained language models also benefit from AutoSelect?
> ---
>
> Thank you for the suggestion. We further conduct experiments on a Qwen3-8B model, which is post trained by reinforcement learning. AutoSelect can still offer considerable performance improvements.
>
>
> | Method \ Task | Winowhy | Epistemic\_reasoning | Aqua | Trec |
> | :--- | :---: | :---: | :---: | :---: |
> | Random                    | 0.617 | 0.793 | 0.424 | 0.364 |
> | Retrieval-based Top-k     | 0.661 | 0.827 | 0.451 | 0.412 |
> | AutoSelect                | 0.703 | 0.897 | 0.475 | 0.483 |

---

> > ### Author Rebuttal · Reviewer_iK2R · 2026-04-03
> >
> > Thank you for your response and the additional experiments. Below, I've included a few more comments regarding your work.
> >
> > - It would be great if you included these new experiments in the paper.
> > - I completely agree that the AutoSelect framework is suitable when the downstream model is fixed. However, the framework is limited to a few-shot setting of moderately sized models. The authors also acknowledge this statement. In my opinion, this is a weakness of the framework. Extending to many-shot settings will make the work more impactful.
> >
> > For these reasons, I will maintain my initial score of 4.

---

> > > ### Author Response · Authors · 2026-04-03
> > >
> > > We sincerely thank the reviewer for the supportive follow-up. We also agree that extending AutoSelect beyond the current fixed-downstream, few-shot setting to many-shot regimes would make the work more impactful.  We have clarified this future direction, as well as potential technical extensions, more explicitly in the revision by incorporating our discussion.

---

### Decision · Program_Chairs · 2026-04-30

**Decision:**

Accept (regular)

**Comment:**

This paper addresses the problem of in-context demonstration selection for few-shot learning with LLMs. The key challenge is that demonstration selection is sensitive to both content (which examples to select) and ordering (the sequence of examples), and these two factors are interdependent. The authors propose AUTOSELECT, an auto-regressive framework that frames demonstration selection as a sequential decision process. Unlike prior work that treats selection as independent retrieval, AUTOSELECT constructs demonstration sequences one step at a time, conditioned on the query and previous selections—similar to how LLMs generate text token by token. The method uses a ViT-based policy model operating on 2D matrix embeddings to preserve token structure, incorporates an adaptive stopping mechanism (EOS signal) to dynamically determine optimal sequence length, and optimizes the policy via a theoretically-grounded cross-entropy objective that minimizes an upper bound on the discrepancy to the optimal Plackett-Luce ranking.

The reviewers note that the problem being studied is meaningful, well-motivated, and the paper is well-written — Figure 2 in particular does a great job of explaining the overall training pipeline (iK2R, 45Mq, pYni). The paper proposes a novel and interesting approach which is theoretically-grounded (iK2R, pYni). The experimental scope being fairly broad (45Mq) and the framework is tractable with efficient inference for demonstrated tasks (SWhi). AutoSelect improves downstream few-shot in-context learning performance, on average, compared to the evaluated baselines (iK2R), with gains on hard tasks are substantial and universal across model sizes (pYni).

During the discussion period, the reviewers also note some weaknesses. iK2R, pYni point to missing baselines (T-Few, RDES) for which authors include new experiments in their rebuttal response and show that AutoSelect outperforms both. Authors also conduct experiments on a Qwen3-8B model, which is post trained by reinforcement learning based on the iK2R’s review and show that AutoSelect can still offer considerable performance improvements for post-trained models as well. SWhi also notes that training of the proposed method requires generating K rollouts per query and evaluating rewards via LLM queries which can be expensive for large scale deployments.

The most common concern raised by three reviewers (iK2R, 45Mq, pYni) is that the framework is limited to a few-shot setting of moderately sized models. The current work will be computationally very expensive for the many-shot ICL which is getting more and more popular these days. The authors acknowledge this weakness and are clear in the paper about the scope and claims of the work — they do not position AutoSelect for extreme many-shot settings, which are closer to long-context prompting / RAG and constitute a significantly different problem setting. The authors are clear that they specifically target the few-shot demonstration selection, where the goal is to maximize accuracy under a limited demonstration budget.

Ultimately, concerns raised by pYni and SWhi are completely addressed during the rebuttal and discussions and recommend acceptance (score 5). iK2R mentions that while there are some clear limitations, given that the paper focuses only on smaller label budgets, they found its approach reasonable and recommend weak acceptance (score 4). Though 45Mq has some reservations, they mention that the discussion has been helpful and has improved their understanding of the paper and its intended scope. They increased their score to 3 (weak reject) stating that the paper’s main results rely primarily on a relatively weak 3B backbone. The authors have some results in the appendix on larger models which were convincing to other reviewers, however 45Mq suggested including at least one main result on a clearly larger model. At the same time, 45Mq also acknowledges that running experiments on larger models impose a substantial hardware or computation burden and if the authors do not have access to such compute, they do not intend this as a strict requirement either. Given all the arguments from reviewers, the clarity in scoping of the paper and sound experimental results for the intended setting, I I lean towards acceptance.